# Mus81-Mms4 endonuclease is an Esc2-STUbL-Cullin8 mitotic substrate impacting on genome integrity

Anja Waizenegger[1], Madhusoodanan Urulangodi [1,4], Carl P. Lehmann [2], Teresa Anne Clarisse Reyes[1], Irene Saugar[2], José Antonio Tercero [2], Barnabas Szakal [1] & Dana Branzei [1,3 ✉]

The Mus81-Mms4 nuclease is activated in G2/M via Mms4 phosphorylation to allow resolution of persistent recombination structures. However, the fate of the activated phosphorylated Mms4 remains unknown. Here we find that Mms4 is engaged by (poly) SUMOylation and ubiquitylation and targeted for proteasome degradation, a process linked to the previously described Mms4 phosphorylation cycle. Mms4 is a mitotic substrate for the SUMO-Targeted Ubiquitin ligase Slx5/8, the SUMO-like domain-containing protein Esc2, and the Mms1-Cul8 ubiquitin ligase. In the absence of these activities, phosphorylated Mms4 accumulates on chromatin in an active state in the next G1, subsequently causing abnormal processing of replication-associated recombination intermediates and delaying the activation of the DNA damage checkpoint. Mus81-Mms4 mutants that stabilize phosphorylated Mms4 have similar detrimental effects on genome integrity. Overall, our findings highlight a replication protection function for Esc2-STUbL-Cul8 and emphasize the importance for genome stability of resetting phosphorylated Mms4 from one cycle to another.

[1] IFOM, the FIRC Institute of Molecular Oncology, Via Adamello 16, 20139 Milan, Italy. [2] Centro de Biología Molecular Severo Ochoa (CSIC/UAM), Cantoblanco 28049 Madrid, Spain. [3] Istituto di Genetica Molecolare, Consiglio Nazionale delle Ricerche (IGM-CNR), Via Abbiategrasso 207, 27100 Pavia, Italy. [4] Present address: Sree Chitra Tirunal Institute for Medical Sciences and Technology (SCTIMST), Thiruvananthapuram, Kerala 695011, India. ✉email: dana.branzei@ifom.eu

Cells need to efficiently process replication-associated recombination structures to prevent chromosome mis-segregation during mitosis. To do so, they rely on the action of structure-specific endonucleases, such as Mus81–Mms4 and Yen1, and on the Sgs1-Top3-Rmi1 (STR) dissolvase complex to process recombination intermediates[1,2]. The STR complex can accurately resolve various recombination intermediates without exchange of information between homologous DNA strands, and its action is manifested from S phase till mitosis[3]. STR is compensated for by the Mus81–Mms4 endonuclease that becomes essential in processing various recombination structures[4–7]. Accordingly, loss of Sgs1 is synthetic lethal with the mms4Δ and mus81Δ mutations, and this synthetic lethality is rescued by deletion of homologous recombination genes[4,8]. Analysis of replication-associated DNA intermediates forming upon genotoxic stress revealed that STR, and not Mus81–Mms4, is required to resolve replication-associated recombination structures[6,7,9,10]. The persistent recombination intermediates accumulating in sgs1Δ mutants upon acute genotoxic stress are largely resolved before anaphase in a manner relying on the Mus81–Mms4 nuclease[6,7], as well as on Cdk1 and Cdc5 kinases[6], which are required along with Dbf4-dependent kinase DDK to activate the Mus81–Mms4 complex in G2/M by phosphorylating Mms4[3,6,11–13].

Because Mus81–Mms4 is a nuclease with activity toward various replication-associated intermediates[14], its functions must be executed in appropriate locations and in the context of adequate protein complexes[3]. A critical activator of Mus81–Mms4 is the G2/M-induced phosphorylation of Mms4[3,6,11–13]. The fate of phosphorylated Mms4 (Mms4-P) is, however, unknown. It could be dephosphorylated and recycled, or alternatively, Mms4 may be degraded and re-synthesized in G1. Here we report a cascade of posttranslational modifications involving SUMOylation and ubiquitylation that accompany the proteasomal degradation of Mms4 and a small pool of its Cdc5 activator in mitosis. Specifically, we determined that the Slx5/8 SUMO-targeted ubiquitin ligase (STUbL), the SUMO-like domain (SLD) containing protein Esc2, and the Cullin8 complex cooperate to maintain the normal cycle of Mms4 phosphorylation. Disruption of these turnover activities leads to abnormal persistence of an active Mms4-Mus81 nuclease in the next G1 phase and throughout the cell cycle. STUbL and Cul8/Mms1 dysfunctions and a stabilized mutant of Mms4-Mus81 associate with abnormal processing of replication-associated recombination intermediates and delayed checkpoint activation in response to genotoxic stress. The results thus uncover a role for Esc2-STUbL-Cul8 in fine-tuning Mus81–Mms4 activity from one cell cycle to another and highlight the importance of regulating the Mms4-P cycle for genome integrity.

## Results

### Mms4 undergoes proteasome-dependent turnover in mitosis.
To study the fate of the activated Mus81–Mms4 nuclease, we used a Tc-HA-MMS4 strain[6] in which Mms4 translation can be inhibited upon addition of tetracycline (Tet)[15]. We synchronized Tc-HA-MMS4 wild-type (WT) cells in G1, released them in the presence or absence of Tet, and after cells completed bulk replication, we added alpha factor (αF) to allow for a new round of synchronization in G1 (see Supplementary Fig. 1 for flow cytometric analysis and assignment of G1 and G2/M peaks). In the absence of Tet, in line with previous reports, Mms4 is cell cycle regulated, with an upshifted form appearing as cells enter G2/M and disappearing as cells progress through mitosis[6,11–13,16] (Fig. 1a). We verified that the upshifted form of Mms4 represents phosphorylated species by subjecting the immunoprecipitated material from G2/M-synchronized WT cells to lambda

phosphatase (λ) treatment in the absence or presence of phosphatase inhibitors (Inh) (Supplementary Fig. 2a). In the presence of Tet, which prevents new Mms4 protein synthesis after release from G1, both forms of Mms4 greatly reduced in amount in G2/M and mitosis, suggestive of protein degradation (Fig. 1a). We further ascertained that Tet addition does not have negative effects on Mms4 modifications/stability by performing a similar experiment in pADH1-HA-MMS4 cells that express MMS4 from an identical promoter with Tc-HA-MMS4 cells except for the lack of the Tet-binding site (Supplementary Fig. 2b). Moreover, inhibiting de novo translation of Mms4 by addition of Tet following acute DNA damage caused similar strong reduction in Mms4 levels in G2/M and mitosis (Supplementary Fig. 2c). Altogether, the results indicate that Mms4 is degraded in G2/M and mitosis, followed by new protein synthesis in G1.

As several proteins are ultimately degraded in a cell cycle-dependent manner by the 26S proteasome, we examined the effect of adding, in addition to Tet, the proteasome inhibitor MG132 to cells that reached G2/M (Fig. 1b). As control for MG132 action, we examined the levels of Cdc5, a known target of the APC/C proteasome system[17]. We found that addition of MG132 stabilized both Mms4 and Cdc5 levels in mitosis (Fig. 1b), supporting the notion that Mms4 is ultimately targeted for proteasome degradation. Moreover, using proteasome-deficient cim3-1 cells at permissive temperatures whereby proteasomal substrates are partially stabilized, we observed stabilization of Mms4 (Fig. 1c). Because Mms4 is bound to chromatin throughout the cell cycle[18] and the ATPase Cdc48/p97 is involved in extracting different proteasome substrates from chromatin[19], we examined a potential involvement of Cdc48 in this process using the temperature-sensitive cdc48-6 allele in conditions permissive for growth that allow cell cycle progression. Both Mms4 and Cdc5 were stabilized in this mutant (Supplementary Fig. 2d). We note that Mms4-P was prominently present in G1 in cdc48-6 and in cim3-1 proteasome mutants (Fig. 1c). This could be explained by defective extraction from chromatin and turnover of Mms4-P. In addition, the persistence of Cdc5 may also contribute to the maintenance of Mms4-P. Thus Mms4 is targeted for proteasome-mediated degradation in mitosis and Cdc48 assists this process.

### STUbL Slx5/8, Esc2, and SUMO chains regulate Mms4 phosphorylation cycle and turnover.
Cdc48 extracts protein complexes that are conjugated with ubiquitin or SUMO modifications from chromatin to facilitate their proteasomal degradation[20]. We examined whether Mms4 undergoes SUMOylation, a post-translational modification reported for mammalian MUS81-EME1 nuclease in response to replication stress or arsenic treatment[21,22]. To this end, we performed a pulldown of all SUMO conjugates under fully denaturing conditions and probed for endogenous Mms4 C-terminally tagged with PK. The employed strains had the endogenous yeast SUMO (Smt3) N-terminally tagged with a 7His-tag ([His]SUMO) so that SUMOylated species are enriched by Ni-NTA pulldown (hereafter referred to as Ni-PD)[23,24]. SUMOylation of Mms4[PK] was specifically detected in cells expressing [His]SUMO and was more abundant in cells arrested in G2/M or released into mitosis (Fig. 2a).

The role of SUMOylation in proteasome-mediated turnover is best understood in the context of the STUbL Slx5/8 family of proteins that can trigger degradation of SUMO-chain-modified substrates[24,25]. Using slx5Δ cells, we observed stabilization of the Mms4-P form throughout the cell cycle, including in mitosis and in G1, although the overall levels of Mms4 still declined (Fig. 2b).

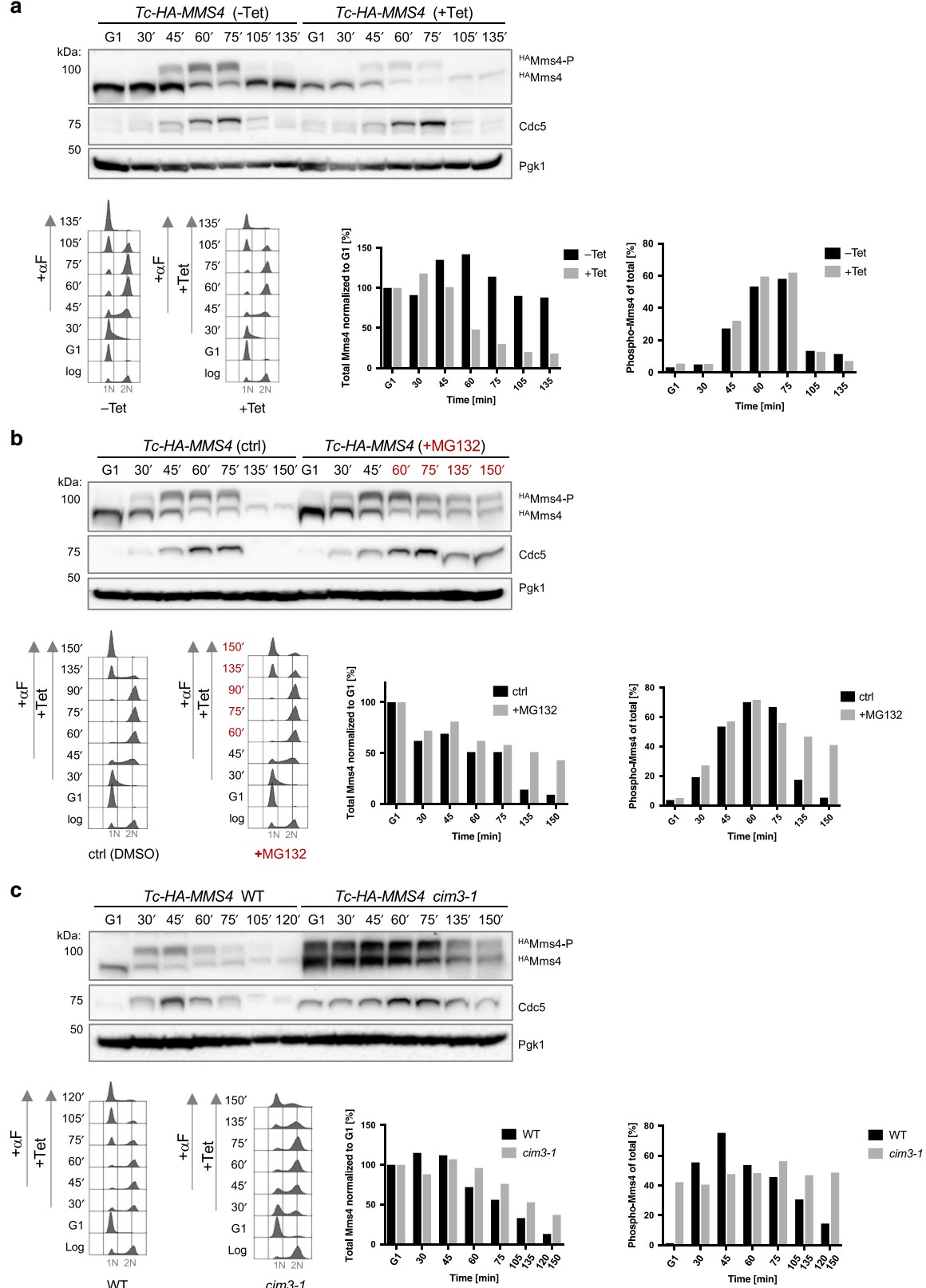

Of interest, Cdc5 was also stabilized in *slx5Δ* mutants, although this level of Cdc5 did not interfere with the ability of cells to exit from mitosis, differently from what is reported for *cdh1Δ* mutants[17]. A similar stabilization of Mms4-P and Cdc5 was observed in cells expressing as single source of SUMO the *smt3-KRall* SUMO variant that cannot form lysine-linked SUMO chains recognized by the STUbL Slx5/8 (Fig. 2c and ref. [24]). Moreover, expressing the *smt3-KRall* SUMO variant as single source of SUMO caused stabilization of low-molecular-weight SUMOylated species of HA-tagged Mms4 in Ni-PD (Fig. 2d). Thus SUMO chains and STUbL engage Mms4 and disrupt the Mms4 phosphorylation cycle.

**Fig. 1 Mms4 undergoes proteasome-dependent turnover in mitosis. a** Time course experiment analyzing Tc-HA-Mms4 protein levels and stability. Logarithmically (log) grown Tc-HA-Mms4 (WT) cells were synchronized in G1 phase with α-factor (αF) and then released in YPD medium in the absence (−Tet) or presence of 1 mM Tetracycline (+Tet). Additionally, after cells reached G2/M (45 min from the initial release), α-factor was added to the culture to arrest cells in the next G1 phase. Samples were taken at the indicated timepoints and the presence of HA-tagged unphosphorylated ($^{HA}$Mms4) and phosphorylated Mms4 ($^{HA}$Mms4-P) species was analyzed by western blot. Cdc5, peaking with Mms4-P in G2/M, was detected using an anti-Cdc5 antibody. Pgk1 served as a loading control. Total levels of Mms4 were quantified by normalization to the loading control and are shown relative to the G1 phase sample. Additionally, the percentage of phosphorylated Mms4 versus total Mms4 is quantified. Cell cycle progression of the cells during the experiment was followed by flow cytometric analysis. 1N and 2N in gray indicate G1 and G2/M phases, respectively. **b** Time course experiment analyzing the turnover of Mms4 in the presence of the proteasome inhibitor MG132. Similar set-up as in **a** using *pdr5Δ* cells to facilitate the uptake of MG132. At 60 min after initial release from G1 arrest, 100 μM MG132 (highlighted in red) or only DMSO (ctrl) was added to the cells, and samples were taken at the indicated timepoints. **c** Time course experiment analyzing Mms4 levels within one cell cycle in the proteasome mutant *cim3-1*. Logarithmically grown cells of the indicated genotype were synchronized in G1 and then released in YPD medium containing 1 mM Tetracycline (Tet) at 30 °C. After cells reached G2/M, α-factor was added to arrest cells in the next G1 phase. Samples for western blot and flow cytometric analysis were taken at the indicated timepoints and processed as in **a**. Source data are provided as a Source data file.

Slx5 interacts with the SLD-containing protein Esc2 and both mediate the turnover of certain proteins, such as Srs2[26,27]. As Esc2 interacts directly with the Mms4-Mus81 endonuclease[28], we next addressed whether Esc2 also participates in regulating the Mms4-P abundance. Similar to *slx5Δ*, in *esc2Δ* cells, Mms4-P and Cdc5 persisted in mitosis and throughout the cell cycle (Fig. 2e). However, because concomitant mutations in Slx5 and Esc2 cause synthetic sickness, it is possible that, in addition to having independent substrates[26], Esc2 affects Mms4-P abundance independently from Slx5.

**Cullin8 complex promotes Mms4 ubiquitylation and regulates the Mms4 phosphorylation cycle and turnover.** Esc2 forms an E3 ubiquitin ligase complex with Cul8 and Mms1[29], an interaction we confirmed using in vivo pulldown between recombinant glutathione *S*-transferase (GST)-Esc2[27] and yeast extracts having endogenous Cul8 and Mms1 tagged C-terminally with the FLAG tag (Supplementary Fig. 3a). We next examined whether Cul8 and Mms1 are involved in regulating Mms4-P abundance. Using a similar approach with the one described above, we found stabilization of Mms4-P and Cdc5 in *cul8Δ* and *mms1Δ* cells (Fig. 3a, b). Moreover, we found similar stabilization of Mms4-P in single *cul8Δ* and *mms1Δ* mutants with that in *esc2Δ cul8Δ* and *esc2Δ mms1Δ* double mutants, although double mutants were slow growing (Supplementary Fig. 3b, c).

As Cul8 and Mms1 comprise the Cullin8 ubiquitin ligase, we next examined whether Mms4 undergoes ubiquitylation. Using Ni-PD in cells expressing *GAL* promoter-inducible His-tagged-ubiquitin gene *UBI4*[30], we found that Mms4 is ubiquitylated, with this modification being strongly reduced in *cul8Δ* mutants but not visibly affected in *esc2Δ* and *slx5Δ* cells (Fig. 3c). We would like to note that our data do not rule out that STUbL ubiquitylates SUMO chain-engaged Mms4, but detection of such SUMO-Ubiquitin chains is technically very challenging as they are prone to accumulate in the wells[24]. Thus STUbL and Cullin8 may separately affect Mms4 turnover and cause stabilization of phosphorylated Mms4 species via different mechanisms. Taken together, our findings indicate that Mms4-P cycle and stability is regulated by Esc2, STUbL, and Cullin8 activities.

**Impaired turnover of Mms4-P leads to increased chromatin association and abnormal Mus81–Mms4 nuclease activity in the next G1 phase.** We next asked whether the observed changes in the Mms4-P cycle affect Mms4 functionality. Mms4 is normally associated with chromatin[18], and thus we asked whether the persistent Mms4-P observed in Esc2-STUbL-Cul8 mutants in G1 (see also Supplementary Fig. 4a) is associated with chromatin or rather present in cytoplasm. Using cell fractionation in cells

arrested in G1, we found increased chromatin association of Mms4 in *slx5Δ*, *esc2Δ*, and *cul8Δ* mutants (Fig. 4a).

Next, we examined whether the observed persistent phosphorylation of Mms4 in G1 associates with an active Mus81–Mms4 nuclease complex, normally limited to G2/M[11,12,16]. To this end, we arrested WT and *esc2Δ*, *slx5Δ*, and *cul8Δ* cells in G1 and G2/M and pulled down the HA-tagged Mms4 proteins from lysates (Fig. 4b and Supplementary Fig. 4b). The nuclease activity was assessed using a $^{32}$P-labeled synthetic 3′-flap containing structure (3′FL), a known substrate for the activated Mms4-Mus81 complex[16]. Whereas for WT cells the nuclease activity of Mus81–Mms4 is detected specifically in extracts of cells arrested in G2/M, in *esc2Δ*, *slx5Δ*, and *cul8Δ* mutants, the nuclease activity of Mus81–Mms4 was detected in both G1 and G2/M extracts (Fig. 4b). Thus individual inactivation of Esc2, STUbL, and Cul8 disrupts the Mms4-P cycle and causes hyperactive Mms4-P on chromatin in G1.

**STUbL and Cullin8 protect replication-associated recombination structures by preventing unscheduled Mms4/Mus81-dependent processing.** To address potential consequences of deregulated Mms4-P during replication, we used two-dimensional (2D) gel electrophoresis to visualize replication-associated recombination intermediates in mutants of STUbL and Cullin8 proximal to origins of replication (Supplementary Fig. 5a). In regard to recombination structure accumulation, single mutants *cul8Δ*, *mms1Δ*, and *slx5Δ* behaved similarly to WT (Supplementary Fig. 5b), indicating that Cul8/Mms1 and Slx5/8 activities are not critical to resolve recombination structures induced by genotoxic stress. This is not surprising as so far only a few factors, namely, subunits of the STR complex and factors affecting its functionality via SUMOylation (Ubc9, the Smc5/6 complex, Esc2)[9,31–35] are critical for this process (see also Supplementary Fig. 5b).

Next, we examined whether Slx5 and Cul8 facilitate instead the formation/stabilization of recombination intermediates accumulating in Sgs1-defective cells. Because of synthetic lethal interactions between *sgs1Δ* and *slx5Δ*[8], and negative fitness interactions between *sgs1Δ* and *cul8Δ*, *mms1Δ*, we used conditional *Tc-sgs1*[36] and *Tc-sgs1-AID* to deplete Sgs1. Cells were synchronized in G1 and then released into S phase in the presence of MMS and Tet, or Tet and Auxin in the case of *Tc-sgs1-AID*, to deplete Sgs1. Replication intermediates were visualized using probes for the early origin of replication ARS305 or the late dormant origin, ARS301. *Tc-sgs1* and *Tc-sgs1-AID* cells depleted of Sgs1 accumulated X-shaped intermediates throughout S phase, similar to results reported for *sgs1Δ*[9] (Fig. 5a, b and Supplementary Figs. 6 and 7). Strikingly, deletion of *MMS1*, *CUL8*, or *SLX5*

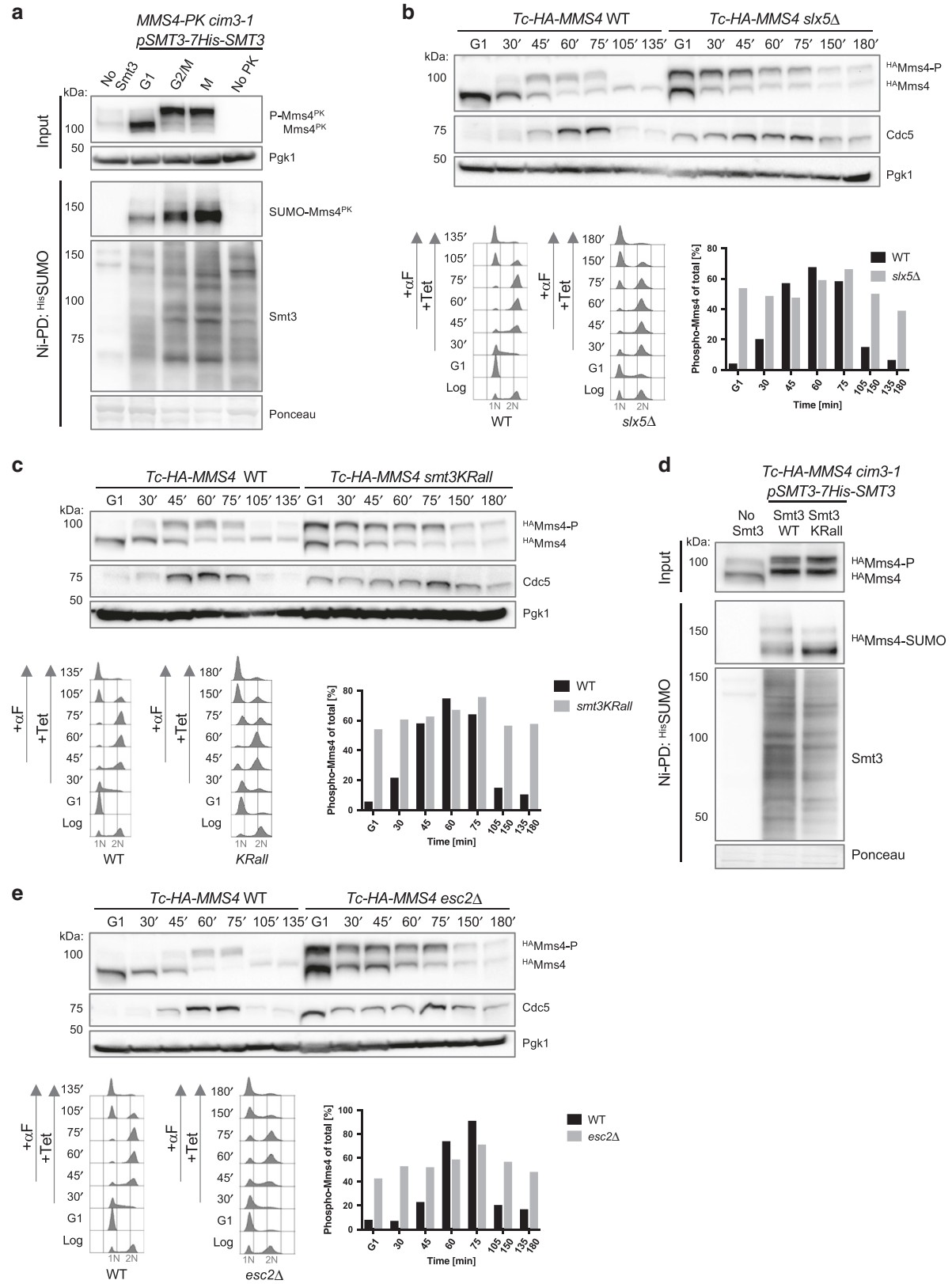

reduced the amount of replication-associated recombination intermediates (Fig. 5a, b and see Supplementary Figs. 6 and 7).

To ask whether the observed reduction in intermediates is caused by abnormal action of Mms4 or reflects a hitherto unknown role of these factors in the formation of recombination intermediates by other means, we examined the effect of mms4Δ

in SLX5-deleted Tc-sgs1 cells. Deletion of MMS4 rescued to a good extent the observed defects (Fig. 5a and Supplementary Fig. 6b). Similar effects were observed in mms1Δ mutants upon either deletion of MMS4 (Fig. 5b) or depletion of Tc-Mms4 (Supplementary Fig. 7b). In the case of Tc-Mms4 depletion, the suppressive effect is primarily visible at late timepoints, which

**Fig. 2 STUbL Slx5/8, Esc2, and SUMO chains regulate Mms4 phosphorylation cycle and turnover. a** Ni-pulldown of His-tagged SUMOylated species from Mms4-PK *cim3-1* cells grown at 30 °C and enriched in G1 (α-factor arrest), G2/M (Nocodazole arrest), and M (G2/M Nocodazole-arrested cells released into mitosis for 40 min). Mms4-PK cells without His-tagged Smt3 (no Smt3) and His-tagged Smt3 cells without Mms4-PK-tag (no PK) were used as controls. Anti-Smt3 antibody was used to detect SUMOylated proteins and anti-PK to detect Mms4 species. Pgk1 and Ponceau staining served as loading controls. **b** Time course experiment analyzing Mms4 species in one cell cycle in WT and *slx5Δ* cells. Logarithmically (log) grown cells were synchronized in G1 phase with α-factor (αF) and then released in YPD medium containing 1 mM Tetracycline (Tet). When cells reached G2/M, α-factor was added again to the culture to arrest cells in the next G1 phase. Samples were taken at the indicated timepoints and the presence of HA-tagged unphosphorylated ($^{HA}$Mms4) or phosphorylated Mms4 ($^{HA}$Mms4-P) and of Cdc5 was analyzed by western blot. Pgk1 served as a loading control. The percentage of phosphorylated Mms4 compared to total levels of Mms4 was quantified. Cell cycle progression of the cells during the experiment was followed by flow cytometric analysis. 1N and 2N in gray indicate G1 and G2/M phases, respectively. **c** Time course experiment analyzing Mms4 species within one cell cycle in cells expressing WT or the lysine-less SUMO variant *smt3-KRall*. Experimental set-up as in **b**. **d** Ni-pulldown of SUMOylated species of HA-Mms4 in cells expressing His-tagged wild-type SUMO or the lysine-less variant *smt3-KRall* from Tc-HA-Mms4 *cim3-1* cells logarithmically grown at 30 °C. Tc-HA-Mms4 cells without His-tagged Smt3 were used as a control (no Smt3). Anti-Smt3 antibody was used to detect SUMOylated proteins and anti-HA to detect Mms4 species. Ponceau staining served as a loading control. **e** Time course experiment analyzing Mms4 species within one cell cycle in WT and *esc2Δ* cells. Experimental set-up as described in **b**. Source data are provided as a Source data file.

correlates to the stabilizing effects of *mms1Δ, cul8Δ* mutations on Mms4 (Supplementary Fig. 7b). Taken together, these results reveal a role for *MMS1, CUL8,* and *SLX5* in protecting replication-associated recombination intermediates against unscheduled Mus81/Mms4-mediated processing.

**Esc2, STUbL, and Cullin8 mutants have delayed DNA damage checkpoint activation.** During replication in the presence of genotoxic stress, the DNA damage checkpoint governed by Mec1-Ddc2 and Rad53 guides error-free outcome of recombination by preventing the premature action of the Mms4-Mus81 endonuclease on the emerging recombination intermediates[6]. To address whether the reduction in replication-associated recombination structures observed in *slx5Δ, cul8Δ* and *mms1Δ* mutants (Fig. 5 and Supplementary Figs. 6 and 7) is associated with checkpoint defects, we examined the kinetics of Rad53 and Rad9 phosphorylation in these backgrounds, using *ddc1Δ* mutants in the checkpoint clamp 9-1-1, previously implicated in replication-associated DNA damage bypass[37], as control. In comparison with WT cells, *esc2Δ, slx5Δ,* and *cul8Δ* manifested delayed activation of Rad53 and reduced Rad9 activation (Fig. 6). Thus defective Esc2-STUbL-Cul8 function interferes with the timely activation of the DNA damage checkpoint.

**Mms4-Mus81 mutant with associated Mms4-P stabilization delays DNA damage response and causes fragility of replication intermediates.** Because Esc2, STUbL, and Cullin8 have multiple substrates, we attempted to identify an Mms4 mutant that may cause similar stabilization of Mms4-P throughout the cell cycle. Esc2 interacts directly with Mus81–Mms4[28] and may bridge STUbL and Cullin8 with which it interacts[27,29] toward Mms4. We used a peptide scan approach to map the interaction domain in Mus81 and Mms4. To this end, we tested interactions of purified GST and GST-Esc2[27] with custom-designed peptide arrays of Mus81–Mms4 15 amino acid (aa) peptides, covering the entire length of these proteins and having peptide–peptide overlaps of 14 aas. The assay highlighted two main response peaks with significantly higher signal-to-noise ratios, one for Mms4 and another for Mus81, with the consensus motifs RSKKSSQVGKLGIKK (Mms4) and EKGTKKRKTRKYIPK (Mus81) (Supplementary Fig. 8a). Due to the significantly higher signal-to-noise ratios of the corresponding peptides, these two responses were considered specific. There are no previously reported domains or interactions within these peptides.

We next generated small internal truncations in Mms4 (Δ541-555) and Mus81 (Δ121-135). The encoded Mus81–Mms4 complex has normal abundance and is still able to interact with

Esc2 by co-immunoprecipitation (co-IP; Supplementary Fig. 8b), suggesting that other Esc2 interaction modules are still present within Mus81 and Mms4 (see Supplementary Fig. 8a), but it causes DNA damage sensitivity (Supplementary Fig. 8c). Notably, however, the internal truncation in Mus81–Mms4 deregulates Mms4-P cycle (Fig. 7a), resembling the effects of *esc2Δ, slx5Δ, cul8Δ* and *mms1Δ* mutants. Of note, the internal truncation in Mms4-Mus81 also caused persistence of Cdc5 without causing defects in mitotic exit (Fig. 7a), indicating that the turnover of a pool of Cdc5, potentially associated with Mms4, is linked to the one of Mms4-P. Notably, the stabilized Mms4-Mus81 mutant with deregulated Mms4-P cycle also delayed Rad53 phosphorylation (Fig. 7b) and caused overall fragility in DNA replication-associated structures (Fig. 7c and Supplementary Fig. 8d). Thus deregulation of the Mms4-P cycle via Mms4 stabilization has negative impacts on genome integrity.

## Discussion
Because of its roles in processing recombination structures that otherwise would impede accurate chromosome segregation, Mus81–Mms4 nuclease is critical for the maintenance of genome stability. We hypothesized that Mus81–Mms4 could cause DNA damage and interfere with normal replication or cell cycle progression if it was present and active at inappropriate times. Therefore, we sought to determine the fate of the phosphorylation-activated Mus81–Mms4 nuclease after its known window of action in G2/M[3,6,11–13,16]. We uncovered that Mms4 is engaged by posttranslational modifications with SUMO and ubiquitin that ultimately trigger its proteasome-mediated degradation in mitosis, in a process involving STUbL Slx5/8, Esc2, and Cullin8. As it is the Mms4-P form primarily persisting in the mutants, it is possible that the active Mus81–Mms4-P complex is the preferential form targeted for degradation. Moreover, our results suggest that Mms4 degradation is coupled to that of a pool of the Cdc5 kinase, the persistence of which allows normal mitotic exit but may contribute to the maintenance of Mms4-P in G1. Finally, we uncover that the deregulated Mms4-P cycle associated with continuous presence of Mms4-P has adverse effects on replication integrity.

Conceptually, protein turnover may be important in the context of nuclear protein quality control or to restrict the levels or activity of a certain substrate. Extensive phosphorylation of Mus81–Mms4 may make accessible buried lysine residues leading to multi, poly-SUMOylation, and/or ubiquitylation and final destruction of the nuclease by the proteasome[38]. In this scenario, inactive, potentially misfolded Mus81–Mms4 nuclease complexes would be targeted for degradation. Conversely, however, we can envisage a process

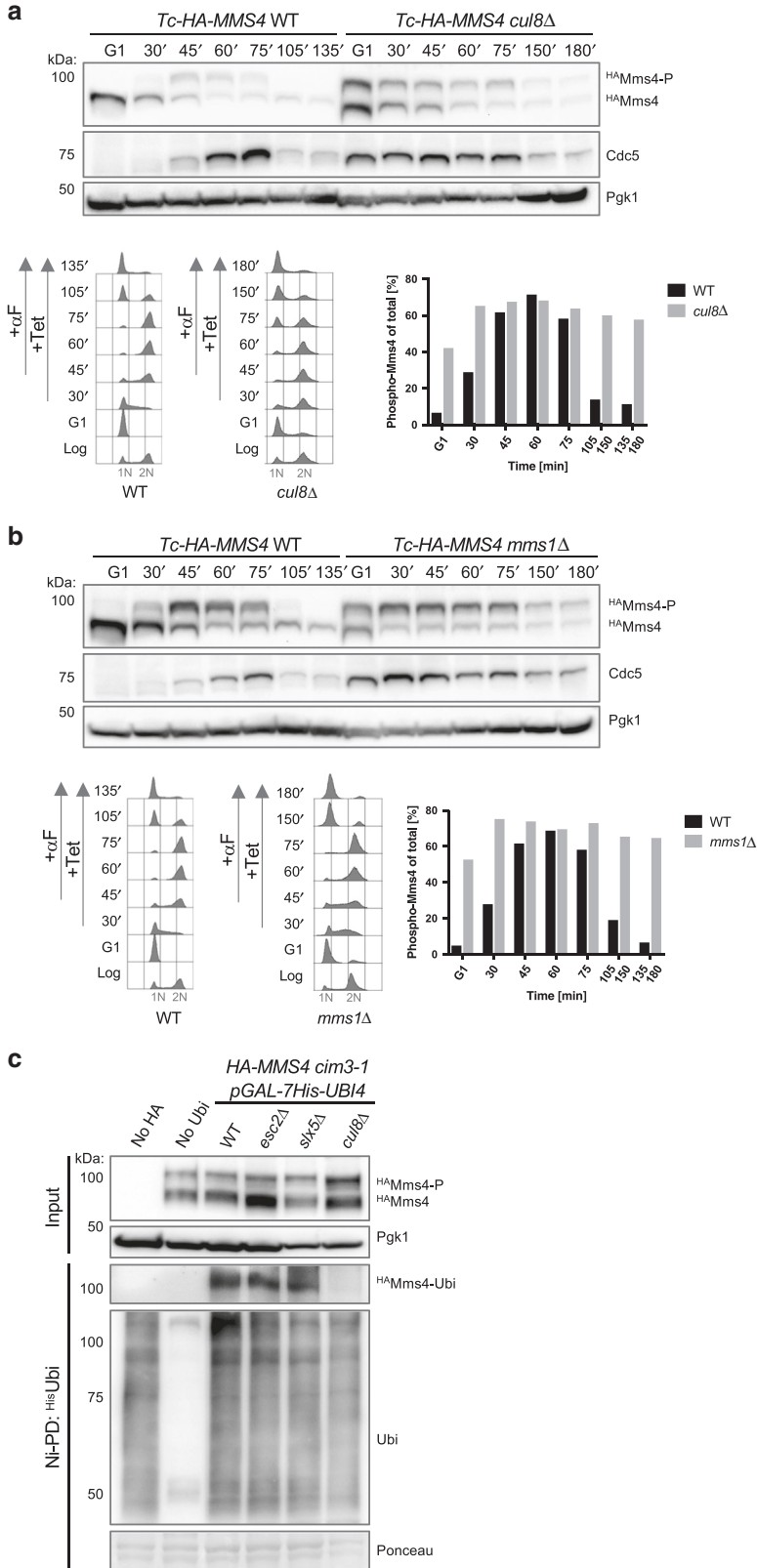

that channels for destruction the phosphorylation-activated Mus81–Mms4 nuclease complexes in order to restrict the inheritance of active Mus81–Mms4 nuclease in the next cell cycle. We find that, in Esc2-Slx5-Cul8 mutants in which Mms4-P is stabilized, there is accumulation of an active nuclease in the next

G1 on chromatin. Thus Mms4 regulation by turnover has in its scope a restriction of the Mus81–Mms4 activity beyond mitosis.

We propose that Esc2 may be a molecular switch in labeling the nuclease molecules to be degraded by recruiting Slx5/8 and Cul8/Mms1 to DNA structures bound by Mus81–Mms4 and

**Fig. 3 Cullin8 complex promotes Mms4 ubiquitylation and regulates the Mms4 phosphorylation cycle and turnover. a** Time course experiment analyzing Mms4 species within one cell cycle in WT and cul8Δ cells. Logarithmically (log) grown Tc-HA-Mms4 cells of the indicated genotype were synchronized in G1 phase with α-factor (αF) and then released in YPD medium containing 1 mM Tetracycline (Tet). After cells reached G2/M, α-factor was added to the culture to arrest cells in the next G1 phase. Samples were taken at the indicated timepoints and the presence of HA-tagged unphosphorylated ([HA]Mms4) or phosphorylated Mms4 ([HA]Mms4-P) species was analyzed by western blot. Immunodetection of Pgk1 served as a loading control. The percentage of phosphorylated Mms4 compared to total levels of Mms4 was quantified. Cell cycle progression of the cells during the experiment was followed by flow cytometric analysis. 1N and 2N in gray indicate G1 and G2/M phases, respectively. Cdc5 levels were detected using anti-Cdc5 antibody. **b** Time course experiment analyzing the Mms4 species within one cell cycle in mms1Δ cells. Experimental set-up as described in **a**. **c** Ni-pulldown of ubiquitylated species of Mms4. His-tagged ubiquitylated proteins ([His]Ubi) were immunoprecipitated from cell lysates using Ni-NTA beads. Proteasome-deficient cim3-1 cells of the indicated genotype (WT, esc2Δ, slx5Δ, cul8Δ) expressing HA-tagged Mms4 were grown in Galactose-containing media to induce the expression of His-tagged ubiquitin. HA-Mms4 cells without His-tagged Ubiquitin (no Ubi) and His-tagged Ubiquitin cells without HA-Mms4 (no HA) were used as controls. Protein samples were analyzed by western blot using an anti-Ubiquitin antibody to detect ubiquitylated proteins and anti-HA to detect unphosphorylated Mms4 ([HA]Mms4), phosphorylated Mms4 ([HA]Mms4-P), and ubiquitylated Mms4 ([HA]Mms4-Ubi). Pgk1 and Ponceau staining served as loading controls. Source data are provided as a Source data file.

Esc2[27,28]. This may happen after Esc2 facilitates Mus81–Mms4 nuclease activity[28] as well as for Mus81–Mms4 complexes that were activated by Cdc5 but their function was not manifested in mitosis.

Why would restriction of Mus81–Mms4 activity be important? Here we find that constantly present Mms4-P throughout the cell cycle results in aberrant processing of replication intermediates and correlates with a delayed checkpoint response. These results thus reveal a hitherto unknown role for Cullin8 and Slx5/8 in replication intermediate protection that is manifested in part via their role in preventing active Mus81–Mms4 complexes to be inherited from the previous cell cycle. An internal truncation mutant in Mus81–Mms4 causing stabilization of Mms4-P also correlates with delayed checkpoint activation and fragility of replication-associated recombination intermediates. Our results indicate that a pool of Cdc5, potentially physically associated with Mms4, is regulated concomitantly with Mms4-Mus81 in mitosis via the Esc2–STUbL–Cul8 axis. This finetuning of Cdc5 and Mms4-P levels does not affect the fitness of cells or their ability to exit mitosis but associates with replication intermediate fragility. Thus our study identifies a crosstalk between the Mus81–Mms4 nuclease cycle and genome integrity orchestrated with the help of SUMO modifiers and ubiquitin ligases to support genome stability.

## Methods

**Antibodies**. As antibodies, anti-FLAG (clone M2, Sigma, #F3165; 1:3000), anti-HA (clone 16B12, BioLegend, #901501; 1:3000), anti-PK (clone SV5-Pk1, AbD Serotec, #MCA1360; 1:3000), anti-Smt3 (clone Y-84, Santa Cruz, #sc28649; 1:2000), anti-Ubiquitin (Abcam, #ab19247; 1:2000), anti-Rad53 (clone EL7, gift from A. Pellicioli; 1:5), anti-Cdc5 (clone yN-19, Santa Cruz, #sc6732; 1:1000), anti-Orc2 (clone SB46, abcam, #ab31930; 1:1000), anti-Pgk1 (clone 22C5D8, Novex Life Technologies, #459250; 1:50.000), anti-Tubulin (clone DM1A, Sigma, #T9026; 1:7000), anti-GST (Cell Signaling, #2622 S; 1:3000), anti-GST Dylight 680 (Rockland, #600-444-200; 1:2000), and anti-HA Dylight 800 (Rockland, #600-445-384; 1:2000) were used.

**Yeast strains, plasmids, and oligonucleotides**. The yeast strains used in this study are derivatives of W303 and the relevant genotypes are shown in Supplementary Table 1. To construct GST-tagged Esc2, the full-length Esc2 gene was amplified from *Saccharomyces cerevisiae* and cloned into the *BamH*I and *Sal*I restriction sites of the pGEX-6P-2 vector, as described below. Oligonucleotides used for 2D gel analysis and nuclease activity assay are listed in Supplementary Table 2.

**Yeast culturing and synchronization**. Unless otherwise indicated, strains were grown at 25 °C in YP-media containing 2% glucose (YPD), as carbon source. For yeast strains harboring a construct with galactose (GAL)-inducible expression, cells were grown overnight in YP-media containing 2% galactose. Cells were synchronized either in G2/M phase by adding nocodazole to a final concentration of 10 μg/ml together with dimethyl sulfoxide (Sigma) to a final concentration of 1%, for 2 h at 28 °C, or in G1 with αF (Genscript) to a final concentration of 3–5 μg/ml for

2–2.5 h at 25 °C. The release from synchronization was performed by washing cells twice in YP media, followed by suspension of cells in YPD media alone or YPD media containing MMS (Sigma) at a final concentration of 0.033 or 0.01%. Following the synchronization step, the experiments were conducted at 28 °C.

To inhibit de novo Mms4 translation in cells expressing *MMS4* from the *pADH1*-Tc-3xHA promoter, Tet was added to a final concentration of 1 mM at the indicated time. For protein depletion by the Auxin-inducible degradation (AID) system, 1 mM Auxin was used. To prevent protein degradation by the proteasome, the proteasome inhibitor MG132 (Sigma) was added to the culture at the indicated time to a final concentration of 100 μM. For MG132 treatment, the *pdr5*Δ background was used to facilitate the uptake.

**Flow cytometry**. For flow cytometric analysis, approximately $1 \times 10^7$ cells for each timepoint were collected and permeabilized in 70% ethanol. Cells were suspended in 10 mM Tris pH 7.5 buffer, and RNA and proteins were removed by RNaseA (0.4 mg/ml) and proteinase K (1 mg/ml) treatment (Sigma). Subsequently, cells were stained in SYTOX green solution (1 μM) (Invitrogen) and analyzed using a FACSCalibur[TM] Flow Cytometer.

**Trichloroacetic acid (TCA) protein extraction**. Briefly, about $1 \times 10^8$ cells were collected, washed with 20% TCA and re-suspended in 200 μl TCA 20%. An equal volume of glass beads was added and the suspension was vortexed for 5 min, following which the extracts were transferred to a fresh tube. The glass beads were washed with 400 μl of TCA 5% and combined with the crude extract. The pellet of proteins was obtained by centrifugation at $845 \times g$ and neutralization with 1 M Tris and resuspension in 100 μl of 2× Laemmli buffer. The samples were boiled for 5 min, centrifuged at $845 \times g$ for 10 min, and the supernatant used for analysis by western blotting.

**Co-immunoprecipitation**. For co-IP analysis of Mms4 and Esc2, cells with HA-tagged Mms4/Mus81 and FLAG-tagged Esc2 were collected (200 ml of $OD_{600} = 1$) and washed once with ice-cold water. Pellets were frozen in liquid nitrogen and subsequently ground to a fine powder with the help of a pre-chilled mortar and pestle. In all, 650 μl of lysis buffer (50 mM HEPES, 50 mM KCl, 0.1% NP40, 1.5 mM MgCl₂, 1.5 mM MnCl₂, 10% Glycerol) were added and the lysate was transferred to an Eppendorf tube. After centrifugation at 4 °C and $10,000 \times g$ for 20 min, the lysate was transferred into a new Eppendorf tube and precleared with 50 μl protein G-Sepharose beads (Thermo Fisher Scientific) for 1 h rotating at 4 °C. Afterwards the lysate was separated from the beads by centrifugation at 4 °C and $10,000 \times g$ for 10 min and the beads were discarded. Then 50 μl precoupled HA-tagged Sepharose beads (Biolegend) were added and incubated with the lysate overnight rotating at 4 °C. The formed complexes of protein, beads, and antibody were then washed three times with lysis buffer before elution in 50 μl of 5× Laemmli buffer (+50 mM dithiothreitol (DTT)). Samples were then boiled for 5 min at 95 °C before western blot analysis. For input, 2% of the lysate was loaded. For the lambda-phosphatase assay, 100 μl of protein G-Sepharose beads (Thermo Fisher) were incubated overnight with anti-HA antibody and yeast native extracts (200 ml of $OD_{600} = 1$) to pull down HA-Mms4. The beads were then washed with PMP buffer (NEB) and for each reaction 25 μl of beads were used as indicated. Samples were mock treated, treated only with lambda phosphatase (NEB) or with lambda phosphatase and phosphatase inhibitors (Phosphatase inhibitor cocktail 2 and 3, Sigma) and incubated for 30 min at 30 °C. Samples were then analyzed by western blotting.

**Ni-pulldown**. For pulldown of SUMOylated or ubiquitylated proteins, yeast cells expressing His-tagged Smt3 ([His]SUMO) or His-tagged Ubiquitin ([His]Ubi) were collected (200 ml of $OD_{600} = 1$) and washed once with ice-cold water. Pellets were

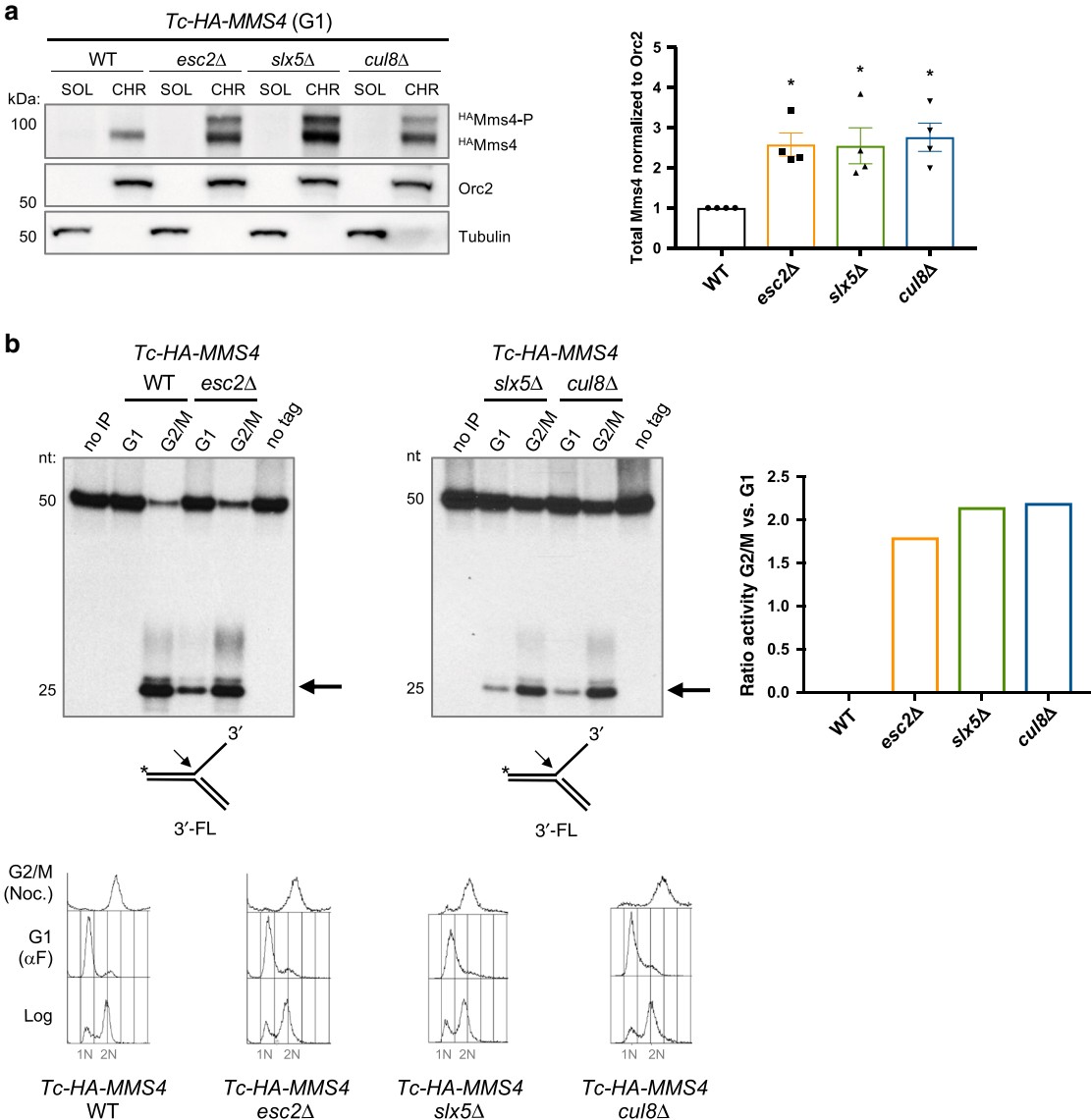

**Fig. 4 Impaired turnover of phosphorylated Mms4 leads to increased chromatin association and abnormal Mus81–Mms4 nuclease activity in the next G1 phase. a** Chromatin binding of Mms4 in G1 phase in WT, *esc2Δ*, *slx5Δ*, and *cul8Δ* cells. Tc-HA-Mms4 cells of the indicated background were synchronized in G1 phase with α-factor and samples were processed to obtain soluble (SOL) and chromatin (CHR) fractions. Both fractions were analyzed by western blot, using an anti-HA antibody. Orc2 served as a chromatin marker and Tubulin as a control for the soluble fraction. The graph represents the quantification of n = 4 independent experiments. Mms4 protein levels of *esc2Δ*, *slx5Δ*, and *cul8Δ* were normalized to Orc2 levels and are shown relative to WT samples. The bars represent the mean values ± SEM. P values were obtained by using a two-tailed unpaired Student's *t* test with Welch's correction, the asterisk indicates *P < 0.05. Compared to WT, P value for *esc2Δ* = 0.012; *slx5Δ* = 0.041; *cul8Δ* = 0.016. **b** Nuclease activity assay of Mms4. Extracts were prepared from Tc-HA-Mms4 WT, *esc2Δ*, *slx5Δ*, and *cul8Δ* cells synchronized in G1 with α-factor or in G2/M with Nocodazole. Mms4 was immunoaffinity purified from the extracts, and the nuclease activity was assessed by the resolution of a $^{32}$P-labeled substrate (3′FL = 3′ flap). The arrow indicates the labeled product resulting from the nucleolytic cleavage of the substrate. As controls, the nuclease assay was either performed with immunoprecipitated extracts from untagged cells (no tag) synchronized in G2/M or the assay was performed in the absence of extract (no IP). Nuclease activity was quantified as the percentage of the cleaved fragment with respect to the total labeled substrate for WT, *esc2Δ*, *slx5Δ*, and *cul8Δ* in G1 and G2/M and then deriving the ratio of activity in G2/M versus G1. Cell synchronization was confirmed by flow cytometric analysis. 1N and 2N in gray indicate G1 and G2/M phases, respectively. nt stands for nucleotide. Source data are provided as a Source data file.

resuspended in 6 ml ice-cold lysis buffer (1.85 M NaOH, 7.5% β-mercaptoethanol), vortexed, and chilled on ice for 15 min. Then 6 ml of TCA 55% were added, mixed, and chilled on ice for 15 min. The lysates were subsequently centrifuged at 4 °C and 1500 × g for 15 min, and the supernatant was discarded. Pellets were washed with 50 ml ice-cold water and centrifuged again at 4 °C and 2500 × g for 5 min. Next, pellets were resuspended in 12 ml Buffer A (6 M guanidine-HCl, 100 mM NaH$_2$PO$_4$, 10 mM Tris-HCl; adjusted pH 8.0 with NaOH; added Tween20 to 0.05% final concentration directly before use), and samples were rotated at room temperature (RT) for at least 1 h. Samples were then transferred to 50 ml Nalgene centrifuge tubes and centrifuged at 4 °C and 23,000 × g for 20 min. The supernatant was collected in a falcon tube and Imidazole was added to a final concentration of 10 mM. Additionally, 50 µl of an Ni-NTA agarose slurry (Qiagen) were added and samples were rotated overnight at 4 °C. Next, samples were washed 3 times with Buffer A and 3 times with Buffer B (8 M urea, 100 mM NaH$_2$PO$_4$, 10 mM Tris-HCl, adjusted to pH 6.3 with HCl; added Tween20 to 0.05% final concentration directly before use). Then samples were centrifuged at 4 °C and 100 × g for 1 min and all liquid was aspirated. Elution of proteins from the beads was achieved by adding 60 µl of HU buffer (8 M urea, 5% sodium dodecyl sulfate (SDS), 200 mM

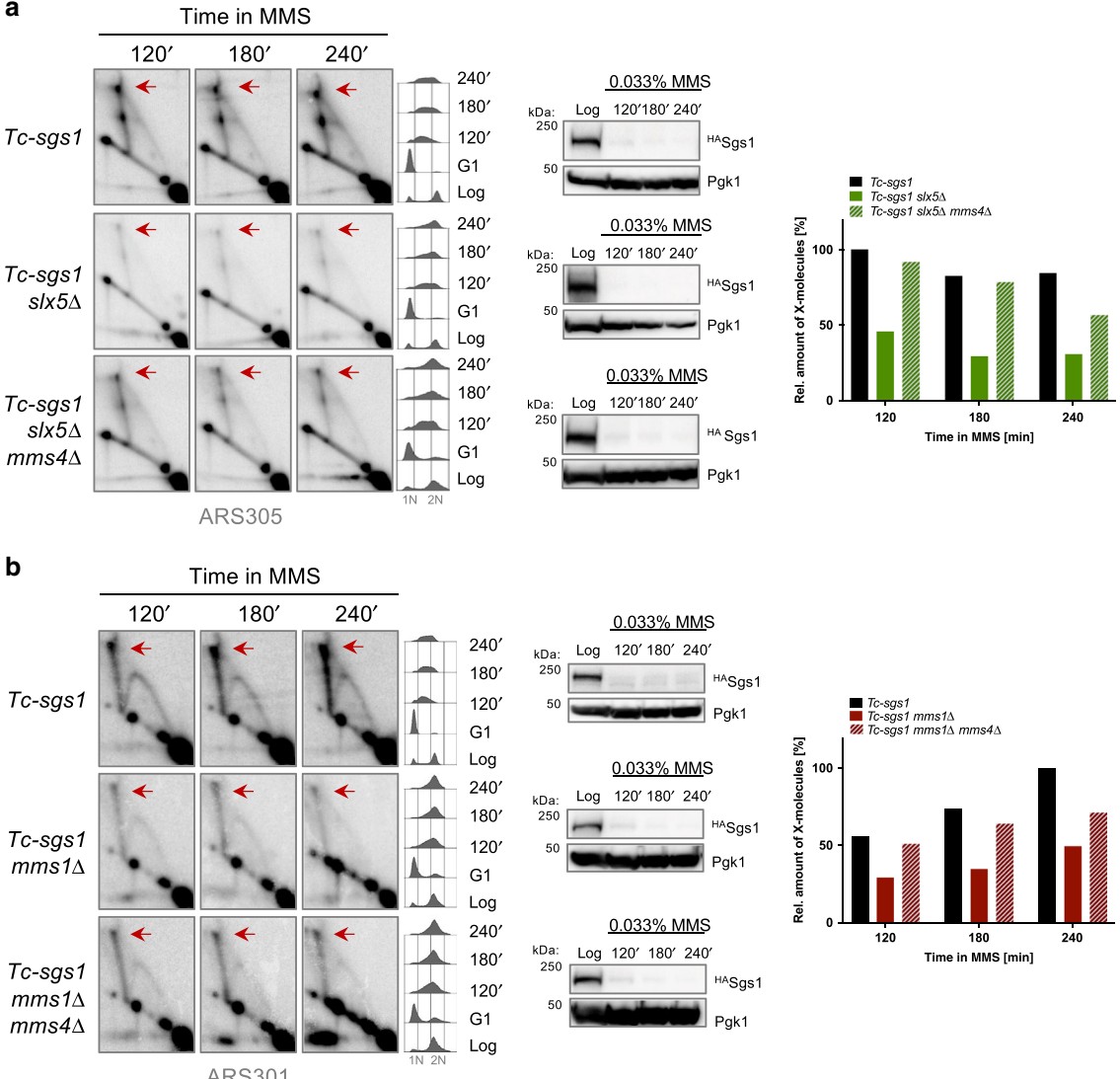

**Fig. 5 STUbL and Cullin8 protect replication-associated recombination structures by preventing unscheduled Mms4/Mus81-dependent processing.**
**a** 2D gel profiles of recombination intermediates isolated from cells of the indicated genotype. Cells were synchronized in G1 phase with α-factor and then released in medium containing 0.033% MMS. Tetracycline 1 mM was added during G1 arrest and release. Samples for 2D gel analysis were collected at the indicated timepoints and levels of HA-tagged Sgs1 were monitored by western blot. Immunodetection of Pgk1 served as a loading control. Cell cycle progression of the cells during the experiment was followed by flow cytometric analysis. 1N and 2N in gray indicate G1 and G2/M phases, respectively. Replication intermediates were digested with *Nco*I and visualized using a radioactively labeled probe specific for ARS305 in 2D gel electrophoresis. Red arrows indicate X-shaped replication intermediates, the relative enrichment of which is shown in the quantified plots. Signal intensities were normalized to the monomer spot and shown relative to the highest value assigned as 100%. **b** 2D gel profiles of recombination intermediates isolated from cells of the indicated genotype. Experimental set-up as in **a**. The replication intermediates were digested with *Eco*RV and *Hind*III and visualized using a radioactively labeled probe specific for ARS301. Source data are provided as a Source data file.

Tris-HCl pH 6.8, 0.05% bromophenol blue, 1.5% DTT) and boiling of the samples at 65 °C for 10 min. Proteins were subsequently analyzed by western blot.

**GST and GST-Esc2 purification.** For GST-pulldown assays, the proteins were purified as described below. *Escherichia coli* BL21 (DE3) cells carrying the pGEX-6P-2-Esc2 plasmid were grown at 37 °C in LB media containing Ampicillin to an $OD_{600} = 0.6$, and protein expression was induced by the addition of 0.5 mM IPTG. After 3 h of induction at 37 °C, the culture was cooled on ice, and the cells were harvested by centrifugation and sonicated in PBST (phosphate-buffered saline containing 1% Triton X-100) containing 10% glycerol and protease inhibitor cocktail (1:100, Calbiochem). The supernatant was collected and incubated with equilibrated glutathione-Sepharose 4B beads (GE Healthcare) at 4 °C for 3 h. The beads were washed twice with PBST buffer and three times with Tris-HCl pH 7.5 buffer containing 500 mM NaCl and eluted with 20 mM reduced glutathione. The eluted proteins were dialyzed overnight at 4 °C against storage buffer (20 mM Tris-

HCl-pH 7.5, 140 mM NaCl, 1 mM DTT, 1 mM EDTA, and 10% glycerol) and divided into aliquots for storage at −80 °C.

**GST in vivo pulldown assays using total yeast cell lysates.** Approximately 5 µg of GST or GST-Esc2 were immobilized on 30 µl of glutathione-Sepharose 4B beads (GE Healthcare). Total cell lysates were prepared from different yeast strains using a solubilizing buffer described below. GST fusion proteins (5 µg) on glutathione beads were incubated with approximately 2.5 mg of yeast cell lysate at 4 °C in Tris-HCl buffer (Tris pH 7.5, 150 mM NaCl, 1 mM DTT, 1 mM EDTA, 10% Glycerol, 0.1% Triton X-100, and Protease inhibitor cocktail 1:100, Calbiochem) for 2–3 h. The beads were washed twice with Tris-HCl buffer and then twice with Tris-Buffer containing NaCl at 350 mM. The protein complexes isolated on the beads were eluted with 30 µl of 2× Laemmli buffer and subjected to 10% SDS–polyacrylamide gel electrophoresis for analysis by western blotting.

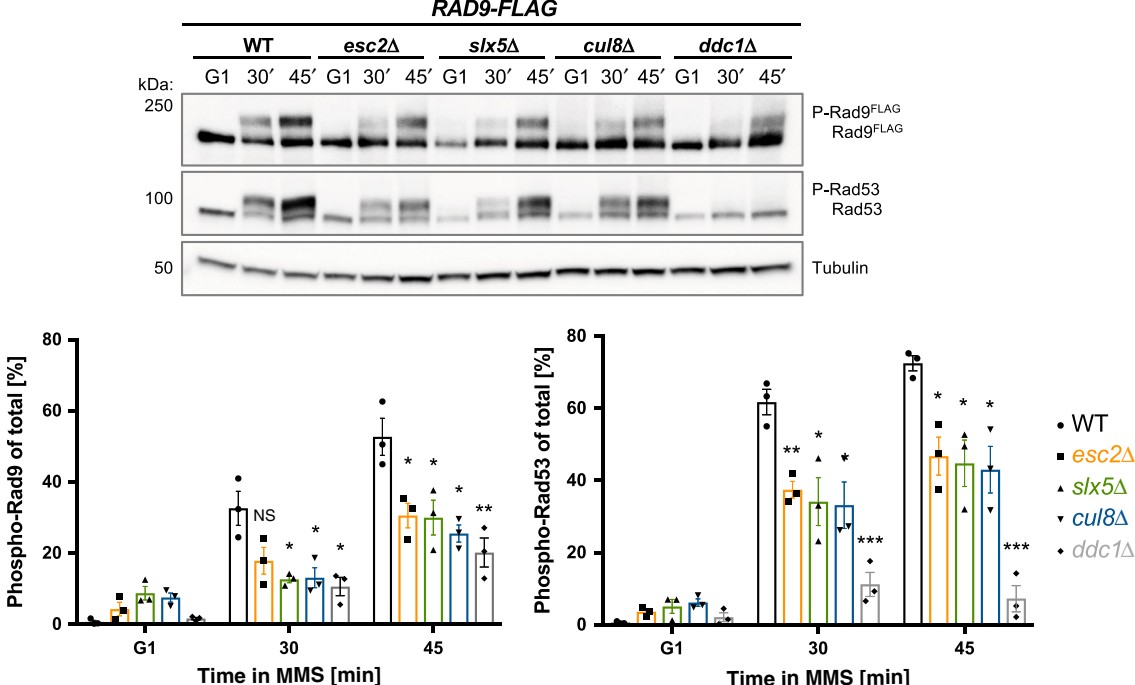

**Fig. 6 Esc2, STUbL, and Cullin8 mutants have delayed DNA damage checkpoint activation.** Kinetics of Rad9 and Rad53 phosphorylation in WT, *esc2*Δ, *slx5*Δ, and *cul8*Δ cells. Rad9-FLAG-expressing cells were synchronized in G1 phase with α-factor and then released in YPD medium containing 0.033% MMS. Samples were taken at the indicated timepoints and prepared for western blot analysis using anti-Rad53 and anti-FLAG antibodies. Immunodetection of Tubulin served as a loading control. Checkpoint-deficient *ddc1*Δ cells were used as control. Three independent experiments (*n* = 3) were used for the quantification of Rad9 and Rad53 phosphorylation; the bars represent the mean values ± SEM. *P* values were obtained by using a two-tailed unpaired Student's *t* test with Welch's correction, the asterisks indicate \**P* < 0.05, \*\**P* < 0.01 and \*\*\**P* < 0.001. *P* values for Rad9 quantification: *esc2*Δ = 0.0302 (45 min); *slx5*Δ = 0.0479 (30 min), 00336 (45 min); *cul8*Δ = 0.0344 (30 min), 0.0206 (45 min); *ddc1*Δ = 0.0259 (30 min), 0.0092 (45 min). *P* values for Rad53 quantification: *esc2*Δ = 0.007 (30 min), 0.0264 (45 min); *slx5*Δ = 0.0343 (30 min), 00392 (45 min); *cul8*Δ = 0.0283 (30 min), 0.0346 (45 min); *ddc1*Δ = 0.0005 (30 min), 0.0004 (45 min). NS stands for not significant. Source data are provided as a Source data file.

**Cell fractionation**. For cell fractionation experiments, approximately $2.5 \times 10^8$ cells were collected and fixed with 1% formaldehyde (Sigma) for 30 min. Cells were washed 3 times with 1× TBS before resuspension in lysis buffer (50 mM Hepes KOH pH 7.5, 140 mM NaCl, 1 mM EDTA, 1% Triton-X100, 0.1% Na-deoxycholate) and subsequent cell disruption using a Multibeads shocker. The lysate was recovered, representing the soluble fraction. The cell pellet was then sonicated in fresh lysis buffer to shear the chromatin. After centrifugation, the supernatant was collected, representing the chromatin fraction. In all, 200 μl of soluble fraction and 200 μl of chromatin fraction were added to the same volume of 2× Laemmli buffer. Both fractions were subsequently analyzed by western blotting.

**2D gel analysis**. Purification of DNA intermediates and the 2D gel procedure were carried out as described in ref. [39,40]. In all, 200 ml cultures ($2 \times 10^9$–$4 \times 10^9$ cells) were arrested by addition of 0.1% sodium azide (final concentration) and cooled down on ice. Cells were harvested by centrifugation, washed in cold water, and incubated in spheroplasting buffer (1 M sorbitol, 100 mM EDTA (pH 8.0), 0.1% β-mercaptoethanol, and 50 U zymolyase/ml) for 1.5 h at 30 °C. In all, 2 ml water, 200 μl RNase A (10 mg/ml), and 2.5 ml Solution I (2% w/v cetyl-trimethyl-ammonium-bromide (CTAB), 1.4 M NaCl, 100 mM Tris–HCl pH 7.6, and 25 mM EDTA pH 8.0) were sequentially added to the spheroplast pellets and samples were incubated for 30 min at 50 °C. In all, 200 μl Proteinase K (20 mg/ml) was then added and the incubation was prolonged at 50 °C for 1 h 30 min and at 30 °C overnight. The sample was then centrifuged at 12,700 × *g* for 10 min. The cellular debris pellet was kept for further extraction, while the supernatant was extracted with 2.5 ml chloroform/isoamylalcohol (24/1) and the DNA in the upper phase was precipitated by addition of 2 volumes Solution II (1% w/v CTAB, 50 mM Tris–HCl (pH 7.6), and 10 mM EDTA) and centrifugation at 3220 × *g* for 10 min. The pellet was resuspended in 2 ml Solution III (1.4 M NaCl, 10 mM Tris–HCl (pH 7.6), and 1 mM EDTA). Residual DNA in the cellular debris pellet was also extracted by resuspension in 2 ml Solution III and incubation at 50 °C for 30 min, followed by extraction in 1 ml chloroform/isoamylalcohol (24/1). The upper phase was pooled together with the main DNA prep. Total DNA was then precipitated with 1 volume isopropanol, washed with 70% ethanol, air dried, and finally resuspended in 1× TE. The DNA samples were digested with *Hind*III and *EcoR*V or *NcoI*, and signals were detected following 2D gel electrophoresis and standard southern blot procedures

using probes against ARS301 (Chr. III 10,135–11,416, see Supplementary Table 2) and ARS305 (Chr. III 39,026–41,647, see Supplementary Table 2). Quantification of 2D gels was performed as described in ref. [40].

**Nuclease activity assay**. The oligonucleotides used to make the DNA 3′FL substrate are listed in Supplementary Table 2. For the formation of the synthetic structures used as the substrates in the nuclease assays, the oligonucleotide RF-1 was 5′-$^{32}$P-labeled using [γ-$^{32}$P]ATP (Perkin Elmer) and T4 Polynucleotide kinase (New England Biolabs) and then annealed with an excess of oligonucleotides. The annealing was performed by heating the DNA molecules for 10 min at 80 °C in 200 mM NaCl plus 60 mM Tris–HCl (pH 7.5) buffer, followed by slow cooling to RT. Tagged-Mms4 was immunoaffinity purified from $7.5 \times 10^8$ cells, which were disrupted using glass beads in 800 μl of binding buffer (see ref. [41]). For HA-Mms4, the supernatant was incubated for 3 h at 4 °C with 12CA5 antibody, followed by 1 h incubation with 15 μl of protein G-Sepharose 4 fast flow (GE Healthcare). The Sepharose-bound proteins were centrifuged, washed extensively, and used directly for the reactions. Nuclease activity assays were based on a previously described method (see ref. [41]). The reaction mixtures (12.5 μl) contained 20 fmol of labeled DNA substrate in 100 mM NaCl, 50 mM Tris–HCl (pH 7.5), 3 mM MgCl$_2$, 250–500 ng poly[dI-dC] plus the immunoaffinity-purified Mms4 protein. The reactions were incubated for 1 h at 30 °C and subsequently the reactions were stopped with denaturing stop buffer (19% formamide, 4 mM EDTA, 0.01% xylene–cyanol, and 0.01% bromophenol). The $^{32}$P-labeled products were analyzed by electrophoresis through 12.5% denaturing gels containing 7 M urea.

**Peptide microarray array**. Peptide microarrays were performed as service by PEPperPRINT GmH in Heidelberg using purified GST-Esc2 and GST versus an array of 15 aa peptides covering the entire Mus81 and Mms4 and having 14 aa overlaps between themselves. The signals were detected using an anti-GST antibody labeled with DyLight680 or a control anti-HA (12CA5) antibody DyLight800. Specifically, the C- and N-termini of Mms4 and Mus81 were elongated and linked by neutral GSGSGSG linkers to avoid truncated peptides. The linked and elongated artificial sequence was translated into 15 aa peptides with a peptide–peptide overlap of 14 aas. The resulting Mms4/Mus81 peptide microarrays contained 1330

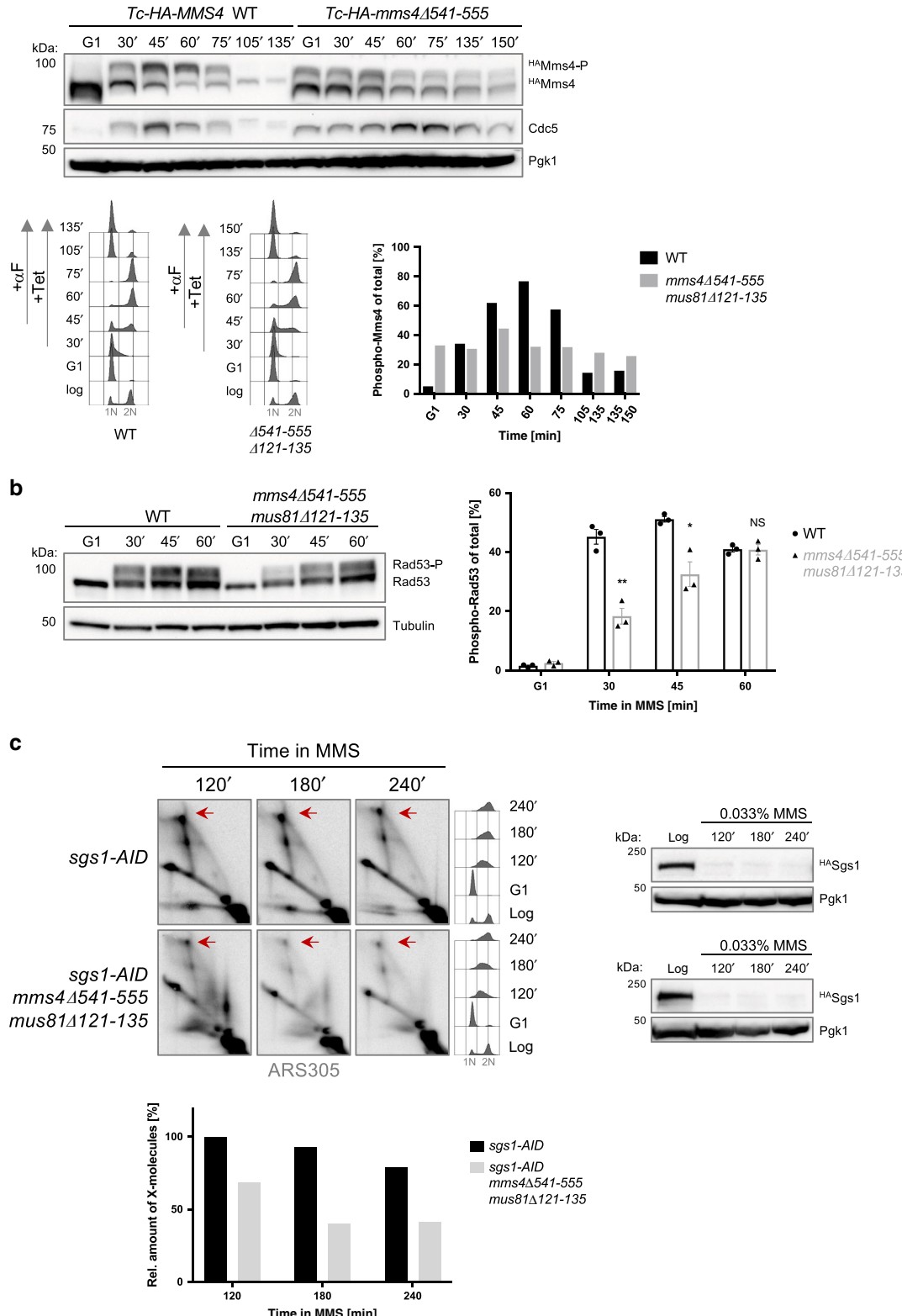

different peptides printed in duplicate (2660 peptide spots) and were framed by HA control peptides (YPYDVPDYAG, 110 spots). The signals were scanned and subsequently quantified using the LI-COR Odyssey Imaging System to generate intensity plots.

**Statistics and reproducibility**. All experiments were repeated at least twice and gave similar results. Experiments in which Mms4 levels were quantified (Fig. 1,

Supplementary Fig. 2, Figs. 2 and 3, Supplementary Fig. 3, and Fig. 7) or the nuclease activity was measured (Fig. 4b) were repeated at least twice independently to confirm the result. For cell fractionation experiments, as well as experiments in which Rad9 or Rad53 levels were measured (Figs. 4a, 6, and 7), three or four experiments were used to allow statistical analyses. The mean values of three or four experiments were calculated and shown with corresponding SEM. Statistical analyses were performed using a two-tailed unpaired *t* test with Welch's correction and the Graphpad Prism software. A *P* value of <0.05 was considered statistically

**Fig. 7 Mms4-Mus81 mutant with associated Mms4-P stabilization delays DNA damage response and causes fragility of replication intermediates.**
**a** Time course experiment analyzing Mms4 species within one cell cycle in WT and *mms4Δ541-555 mus81Δ121-135*. Logarithmically (log) grown cells expressing Mms4 and Mus81 variants from *Tc-ADH1-HA* promoters were synchronized in G1 phase with α-factor (αF) and then released in YPD medium in the presence of 1 mM Tetracycline (+Tet). After cells reached G2/M, α-factor was again added to the culture. Samples were taken at the indicated timepoints and the presence of HA-tagged Mms4 species and of Cdc5 was analyzed by western blot. Pgk1 served as loading control. The percentage of phosphorylated Mms4 versus total Mms4 was quantified. Cell cycle progression of the cells during the experiment was followed by flow cytometric analysis. 1N and 2N in gray indicate G1 and G2/M phases, respectively. **b** Kinetics of Rad53 phosphorylation in WT and the *mms4Δ541-555 mus81Δ121-135* truncation mutant (see **a**). Cells were synchronized in G1 phase and then released in YPD medium containing 0.033% MMS. Samples were taken at the indicated timepoints and analyzed by western blot for Rad53 and for Tubulin, which served as loading control. Three independent experiments ($n = 3$) were used for the quantification of Rad53 phosphorylation; the bars represent the mean values ± SEM. $P$ values were obtained by using a two-tailed unpaired Student's $t$ test with Welch's correction, the asterisks indicate *$P < 0.05$ and **$P < 0.01$. $P$ value for 30 min = 0.0018, 45 min = 0.0417. NS stands for not significant. **c** 2D gel profiles of recombination intermediates isolated from cells of the indicated genotype. *SGS1-AID* and *mms4, mus81* alleles are expressed from *Tc-ADH1-HA* promoters, not indicated. Cells were synchronized in G1 phase with α-factor and then released in medium containing 0.033% MMS. Auxin was added during G1 arrest and after release to induce Sgs1-AID depletion, monitored by western blot. Pgk1 served as loading control. Cell cycle progression of the cells during the experiment was followed by flow cytometric analysis. Replication intermediates were digested with NcoI and visualized using a radioactively labeled probe specific for ARS305 in 2D gel electrophoresis. Signal intensities were quantified, normalized to the monomer spot, and shown relative to the highest value. Red arrows indicate X-shaped replication intermediates. Source data are provided as a Source data file.

significant. Pulldown experiments (Figs. 2 and 3 and Supplementary Figs. 3 and 8) were repeated at least twice and one representative western blot was selected for the figure. 2D gel experiments were repeated at least twice and, in some cases, additionally different alleles or regions were used to confirm the results.

**Reporting summary**. Further information on research design is available in the Nature Research Reporting Summary linked to this article.

## Data availability
The authors declare that all data supporting the findings of this study are available within the paper and its supplementary information files. All data are available from the authors upon reasonable request. Source data are provided with this paper.

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

## Acknowledgements

We thank all Branzei laboratory members for discussions, I. Psakhye and C. R. Joseph for technical help, and R. Visintin for sharing technical information. We acknowledge the service from PERperPRINT GmbH in Heidelberger for the peptide scan array service. This work was supported by the Italian Association for Cancer Research (IG 18976 and IG23710) and European Research Council (Consolidator Grant 682190) grants to D.B. and by the Spanish Ministry of Science, Innovation and Universities (BFU2016-77663-P AEI/FEDER UE) to J.A.T.

## Author contributions

A.W., M.U., B.S., and D.B. designed the research; A.W., M.U., C.P.L., T.A.C.R., I.S., J.A.T., and B.S. performed the experiments; A.W., J.A.T., and D.B. analyzed the data; A.W., B.S., J.A.T., and D.B. made the figures; D.B. wrote the paper; and all authors contributed suggestions.

## Competing interests

The authors declare no competing interests.
