## [Peer Review File · Nature Communications]

Reviewers' comments:

Reviewer #1 (Remarks to the Author):

In the manuscript "Mus81-Mms4 endonuclease is an Esc2-STUbL-1 Cullin8 mitotic substrate tuning the DNA damage response" Dana Branzei and coworkers investigate the regulation of MUS81-MMS4, an endonuclease involved in the processing of DNA structures such as stalled replication forks and Holliday-Junctions. The MMS4 regulatory subunit is known from previous work to be controlled in a cell cycle-dependent manner leading to up-regulation of its catalytic activity in G2 and M phase of the cell cycle, which appears to be critical to avoid hyperactive MUS81-MMS4 nuclease during DNA replication. Here the authors investigate, how the MUS81-MMS4 becomes inactivated after M phase. This is clearly an interesting question that has previously not been addressed.

This manuscript can be divided into two parts: the first showing that phosphorylated Mms4 is degraded in/after M phase by a STUbL-dependent mechanism, the second attempting to show a biological relevance of this degradation. Experiments in both parts have their individual shortcomings as indicated in my points below. I think the authors should be able to fix the problems with the first part relatively straightforwardly. The concerns for the second part are even more severe and I think the authors will require a stabilizing MMS4 mutant to address my concerns definitively. Going forward, I would therefore suggest the authors to consider whether leaving out this premature part of the paper may be an option. This comes with the problem that simply showing the regulation of MMS4 without a physiological role might not be enough for the broad readership of Nature Communications, but perhaps additional impact can be generated by showing evolutionary conservation.

Main points:

1. Fig. 1 – the authors need to find a way to present quantifications of phosphorylated, non-phosphorylated and total MMS4 species. Also, the data needs to be controlled to wildtype cells, best on the same blot. We need to be able to compare effects between different figure panels, which currently we cannot.
2. Fig. 1 – the representation of experiments using often very different timepoints (for example Fig 1A 30, 45, 60, 75, 105, 135 min after G1 <-> Fig 1C 60, 75, 90, 120, 150, 180) makes comparison difficult and non-intuitive. I understand that different rates of proliferation necessitate in some cases later time-points, but for sake of comparison the earlier timepoints are also important.
3. Fig. 1 – I think the combination of promoter shut-off and synchronous cell cycle release make interpretation of these experiments very difficult. As the authors speculate that it is specifically the phosphorylated version of MMS4, it should be straightforward to make a simple experiment where one arrests cells in M phase and then shuts off transcription (or overall translation with CHX) and looks at degradation of MMS4.
4. Fig. 1, but all figures; information in the figure legends is too sparse and sometimes contradicting what is written in the text. For example: - Temperature of CDC48 experiment (just mentioned in text line 114-115); - Fig 1C: figure legends state that MG132 was added together with Tet after G1 release, text (line 101-102) states it was added 1h after release from G1
5. Fig. 1b: The proteasome mutant seems to show much less phosphorylated MMS4 (if comparable with Fig 1a and the others), which is inconsistent with proteasomal degradation and the experiment with proteasome inhibition.
6. Fig. 1c: Phosphorylated form seems not very much stabilized by proteasome inhibition.
7. Fig. 1d: Again stabilization is not very visual. Is this just because of the use of different

timepoints? I cannot judge whether the data is in support of the authors model, but if it was they are doing themselves a disfavor by the presentation.

8. Fig. 2: In my eyes the clearest part of the paper. The stabilization of the phosphorylated version in G1 cells is believable. However, if they want to argue that all these factors act in one pathway, they need to show us double mutants and that those do not further increase the effect on degradation.

9. Fig. 2a: An additional control using MMS4 without tag would be necessary.

10. Fig. 3a: They show enhanced levels of MMS4 on chromatin in the absence of SLX5 etc, but most likely this is simply due to enhanced levels of MMS4. Therefore showing total levels of MMS4 in G1 will be meaningful.

11. Fig. 4 and 5: To assess what is the function of the described degradation mechanism it will be necessary to find a specific mutant in MMS4 that is deficient in this degradative mechanism (for example: SUMO-site mutant, ubiquitination-site mutant, interaction-deficient point mutant). Otherwise, the authors have to use rather pleiotropic mutants. The rescue strategy with the promotor shut-off of MMS4 is not always convincing.

12. Fig. 4: The TC-MMS4 mutant relies on MMS4 degradation to work. Is it really a good strategy to use this mutant in the background of mutants that presumably are degradation deficient?

13. Fig. 4c: The authors show quantification of a single experiment and some of the effects appear to be small (and not easily visible on the gels itself; exception perhaps 240 min timepoint in Fig. 4b). How do we know they are meaningful/significant? Can the authors average over different experiments? Can the authors at least show that other experiments had the same trends?

14. Fig. 5a: How is the quantification done? How is it possible that Rad53 after 45min is only 35% phosphorylated. This number does not match the optical impression.

15. Fig. 5b: I am not convinced that the Mms4 over-expression scenario is analogous to the other mutant scenarios. How do the levels of Mus81 change if the authors inhibit degradation or overexpress Mms4.

16. Fig. 5b: Do the authors really want to concentrate purely on levels. Is it not more promising to focus on the presence of hyper-phosphorylated MMS4 G1 or S. In that vein, can they rescue this phenotype by using previously characterized phosphorylation-deficient mutants?

Reviewer #2 (Remarks to the Author):

'Mus81-Mms4 endonuclease is an Esc2-STUbL- 1 Cullin8 mitotic substrate tuning the DNA damage response'

In this study Waizenegger et al. report that phosphorylated Mms4 is channelled towards Cdc48 segregase-mediated proteasomal degradation during mitosis, in a process involving the SUMO-targeted Ubiquitin ligase Slx5/8, Esc2 and Mms1-Cul8. Moreover they provide evidence suggesting that mitotic degradation of phosphorylated Mms4 avoids accumulation of active Mus81-Mms4 on chromatin during the subsequent S-phase, which would otherwise cause abnormal processing of recombination intermediates and delay activation of the checkpoint.

In my opinion, the work/model is very interesting and the findings merit publication in a broad journal such as Nature Communications. I would however suggest that the authors address the following aspects of their data, which in my opinion will help strengthen their model.

Figure 1:

- The authors never formally show that the Tet-mediated inhibition of Mms4 translation actually works. To be able to claim it in the text, the authors should perform an experiment in which they omit Tet (or even better add Tet to a WT MMS4 strain). Otherwise it is unclear to me that the small decrease in Mms4 abundance at later timepoints of the cell cycle actually comes from the inhibition of translation
- In Figure 1, but also in other Figures, Tubulin is clearly not a good loading control. It is obvious that its abundance is cell cycle regulated, which makes it difficult to then claim that levels of Mms4 fluctuate in a specific manner. Tubulin could be used if the authors loaded two conditions (e.g. +/- Tet) in the same gel – if in both timecourses tubulin looks identical, then Tet-mediated changes in Mms4 levels could be inferred.
- Mms4 levels should be quantified, as previously done by the authors in other publications (e.g. Szakal 2013).
- Inhibition of the proteasome is known to delay cell cycle progression, for example by preventing the degradation of cyclins. Thus, the delayed degradation of Mms4 can be a consequence of cells getting stuck at G2/M. In essence, the authors should be careful describing their data as it does not formally show that Mms4 is a proteasomal substrate.
- All the findings in panels B, C and D could be explained if Cdc5 would not be efficiently degraded in proteasome or cdc48 mutants. The authors should probe their samples for Cdc5. This is particularly important because Cdc5 is known to be a target of the APC/proteasome system.
- The authors describe Figure 1D as having normal kinetics of Mms4 phosphorylation. This is not the case as Mms4 is modified at least 15 minutes faster than in the WT.

Figure 2:

- I am not sure the authors can formally claim that Mms4 is targeted by multiple Smt3 molecules. In panel A, the shifts could be caused by differential phosphorylation of sumoylated Mms4. The authors could soften their interpretation of the data or do an experiment where they phosphatase-treat the Ni-PD of Smt3.
- Treatment with Nocodazole leads to a cell cycle arrest due to activation of the SAC, which is technically an M-phase arrest. Authors refer to it as G2, which is imprecise.
- In panels B-E, authors should probe their WBs against Cdc5. It could easily be that Cdc5 is stabilised in the mutants analysed and lead to persistence of modified Mms4.
- It is very obvious that Mms4 protein levels still decrease significantly in the mutants analysed. How do the authors explain this? It would be important to load the mutants analysed against the WT in the same gel and quantify Mms4 levels.

Figure 3:

- Even though the data looks good, the Mus81-Mms4 activity assays could be quantified and the activity normalised to the protein in the IP.

Figure 4:

- This data is overall very nice, but there is one aspect which I believe should be looked into. Even though Sgs1 levels are not quantified in the Wbs, which could be done, it seems to me that its depletion is not comparable in all conditions. My impression is that Sgs1 levels are always higher when joint molecules are lower, which, if true, would compromise some conclusion made. This is a critical experiment and I think authors should load all the conditions Vs. the WT and quantify carefully Sgs1 levels from the same gel. Otherwise the interpretation of the data could be ambiguous.

Figure 5:

- The authors should treat WT and GAL-MMS4 cells with galactose and perform the same analysis. Comparing cells growing in two different carbon sources is not ideal as the growth rates and overall physiology of the cells will be different. I don't know it for certain, but I would not be surprised if the effect observed was due to the Galactose vs Raffinose rather than the over-

expression of Mms4.

Figure 6:

- I like the first part of the model in panel A, but I don't think panel B is helpful. I don't think it summarises any of the important findings made in the paper.

Reviewer #3 (Remarks to the Author):

This manuscript by Waizenegger et al describes an interesting analysis on Mms4 turn-over as required for proper chromosome segregation. The author's finding that MMS4 is affected by both Slx5/8 and Mms1/Cul ubiquitin ligases is interesting and new. Mms4 and Mms4-P were stabilized in *slx5Δ*, *esc2Δ*, and *cul8Δ* strains. Following up on earlier suggestive data, the authors confirm that the Mms4 and Mms4-P are degraded in a cell-cycle dependent manner. This degradation can be prevented (or reduced) when proteasome activity is prevented. Additionally, Mms4 and Mms4-P are stabilized in a *cdc48* mutant which apparently also fails to extract Mms4 from chromatin. Proof that Mms4 is sumoylated in Fig 2a is marginal and ubiquitylation was not shown directly. I am not clear how the results in Fig. 4 reveal a role for Slx5/8 and Mms1/Cul ubiquitin ligases in replication-associated recombination intermediate stability by preventing abnormal Mms4-Mus81 action associated with defective turnover. The authors need to either discuss the data in more detail or provide additional data to drive this point home. Finally, data on Rad9 and Rad53 phosphorylation suggests that DNA damage checkpoints are not properly engaged if MMS4 turn-over is defective. However, differences in Rad9 and Rad53 phosphorylation between WT and mutants seem minimal (at least by visual inspections) and there are questions how the quantitation was performed. Overall, this is a well-written and interesting story that seeks to establish the interplay of PTMs of Mms4 and protein degradation machinery to prevent its inappropriate activity in the G1 stage of the cell cycle.

MAJOR:

Proof that MMS4 is sumoylated in Fig 2a is marginal. MMS4-PK was co-expressed with SMT3-HIS6. After ppt of MMS4PK the authors detect SUMO-HIS6 in the pull-down lanes. This could be improved with a SUMO-shift assay with epitope-tagged and untagged SUMO. Likewise, ubiquitylation of Mms4 in WT and ubiquitin ligase mutants should be shown in the main document.

Fig 5: This is a western blot film that appears to be scanned for quantitation. Since the authors are quantitating minute differences in protein levels and because western blot signals on film are non-linear, this technique is not appropriate for data quantitation. The authors should use an Odyssey or similar system to quantitate their western blots. The description how the data was analyzed is at a minimum (e.g. what were the bands normalized to?). Additionally, taking into account that levels of the loading control as well as the G1 levels vary, there is reduced confidence in these data. Finally, WT and *slx5Δ* (and possibly other strains) show vastly different growth rates and so a comparison of checkpoint protein activation is difficult at best. Fig 5b quantitation has the same problems with the additional caveat that GAL overexpression phenotypes are very difficult to interpret.

MINOR:

In Fig 1a (Tc-HA-MMS4(WT) ... there seems to be another prominent band (partially cut of) in the lane that runs above MMS4-P. The authors should show more of the blot and comment on the identity of this band.

- line 160-161 ... the authors indicate that MMS4-P is stabilized ... but so is the non P form of MMS4! This should be noted and discussed.
 - The role of SUMO in degradation of MMS4 is unclear as both MMS4 and MMS4-P are degraded and stabilized. The authors should discuss this.
 - Fig 3: Enhanced chromatin association of MMS4 in *esc2*, *slx5Δ* *cul8Δ* is beautiful. I am wondering if tubulin is the best control though.
 - Line 182: Thus, defective Mms4-P turnover causes persistent high activity of the Mus81-Mms4 nuclease on chromatin. ... this was only shown for *esc2Δ* (3b) ... does it behave the same in *slx5Δ* or *cul8Δ*? This is important to assess the role that sumoylation and STUbLs play in this process, ... is the figure mislabeled? Both left and right figures are labeled Tc-HA-MMS4 WT *esc2Δ*
 - Line: 205 cells depleted for Sgs1 accumulated X-shaped intermediates ... please indicate (e.g. using an arrow) these features. What is being quantitated here?
 - Along the same line, I am not clear how the results in Fig. 4 reveal a role for MMS1, CUL8 and SLX5 in replication-associated recombination intermediate stability by preventing abnormal Mms4-Mus81 action associated with defective turnover. The authors need to either discuss the data in more detail or provide additional data (and controls) to drive home this point.
- Line 255: We found that Mms4-P is engaged by posttranslational modifications with SUMO and ubiquitin ... BUT no ubiquitin is shown in main figures of the ms.
- Line 266: Extensive phosphorylation of Mus81-Mms4 may cause its unfolding, exposing buried lysine residues leading to multi, poly-SUMOylation ... Unfolding is very unlikely, maybe just state "make accessible"?

Reviewers' comments:

Reviewer #1 (Remarks to the Author):

In the manuscript “Mus81-Mms4 endonuclease is an Esc2-STUbL- 1 Cullin8 mitotic substrate tuning the DNA damage response” Dana Brnzei and coworkers investigate the regulation of MUS81-MMS4, an endonuclease involved in the processing of DNA structures such as stalled replication forks and Holliday-Junctions. The MMS4 regulatory subunit is known from previous work to be controlled in a cell cycle-dependent manner leading to up-regulation of its catalytic activity in G2 and M phase of the cell cycle, which appears to be critical to avoid hyperactive MUS81-MMS4 nuclease during DNA replication. Here the authors investigate, how the MUS81-MMS4 becomes inactivated after M phase. This is clearly an interesting question that has previously not been addressed.

This manuscript can be divided into two parts: the first showing that phosphorylated Mms4 is degraded in/after M phase by a STUbL-dependent mechanism, the second attempting to show a biological relevance of this degradation. Experiments in both parts have their individual shortcomings as indicated in my points below. I think the authors should be able to fix the problems with the first part relatively straightforwardly. The concerns for the second part are even more severe and I think the authors will require a stabilizing MMS4 mutant to address my concerns definitively. Going forward, I would therefore suggest the authors to consider whether leaving out this premature part of the paper may be an option. This comes with the problem that simply showing the regulation of MMS4 without a physiological role might not be enough for the broad readership of Nature Communications, but perhaps additional impact can be generated by showing evolutionary conservation.

We thank the reviewer for finding our work of interest and for suggesting specific ways to improve the paper. We have now addressed the criticisms in both parts. Importantly, we also succeeded in identifying a stabilized Mus81-Mms4 mutant that resembles STUbL and Cul8/Mms1 mutants in regard to the effect on checkpoint activation and hyperactivity on DNA replication-associated recombination structures.

Main points:

1. Fig. 1 – the authors need to find a way to present quantifications of phosphorylated, non-phosphorylated and total MMS4 species. Also, the data needs to be controlled to wildtype cells, best on the same blot. We need to be able to compare effects between different figure panels, which currently we cannot.

We have repeated the experiments to load wt and mutants in the same blot and provide quantifications of P-Mms4 versus total Mms4 for different experimental conditions/mutants and, where appropriate, of Mms4 levels versus the initial start-point of total Mms4 in G1 to assess degradation/loss. We believe that the comparisons are straightforward now.

2. Fig. 1 – the representation of experiments using often very different timepoints (for example Fig 1A 30, 45, 60, 75, 105,135 min after G1 <-> Fig 1C 60, 75, 90,120, 150, 180) makes comparison difficult and non-intuitive. I understand that different rates of proliferation necessitate in some cases later time-points, but for sake of comparison the earlier timepoints are also important.

As mentioned above, we have now loaded control samples in the same gel to facilitate comparison. For the example given by the reviewer, we note that these were samples treated or not with MMS. For straightforward comparison when dealing with MMS, we

have performed a new experiment and loaded samples in the same conditions of acute MMS treatment, in the absence or presence of Tetracycline (Supplementary Fig. 1c). The clear result from these experiments is that Mms4 and its phosphorylated form are getting degraded during mitosis, irrespective of whether cells experienced prior DNA damage in S phase (Figure 1a and Supplementary Fig. 1c). The observed degradation is impaired in proteasome mutants or upon proteasome inhibition with MG132 (Figures 1b, c).

We further like to note that in most cases we have used the same time points in both parts of the experiments (control vs mutant/condition/inhibitor). However, in some cases we considered more informative – considering also that the number of samples we can load in the same gel is limited – to use slightly different time points at the last stages of the experiment. This is because in these cases cells do not progress equally well and the chosen time points compare more precisely similar stages of the cell cycle. For example, Figure 1c and Supplementary Fig. 1d: WT105' is compared with *cim3-1* and *cdc48-6* 135', as there is a delay in the mutants, whereas same times (i.e. 105') would not be showing the same stage.

3. Fig. 1 – I think the combination of promotor shut-off and synchronous cell cycle release make interpretation of these experiments very difficult. As the authors speculate that it is specifically the phosphorylated version of MMS4, it should be straightforward to make a simple experiment where one arrests cells in M phase and then shuts off transcription (or overall translation with CHX) and looks at degradation of MMS4.

We concluded that both forms of Mms4 are degraded (Figure 1a) and speculated that the turnover may be targeting initially the phosphorylated form that is induced in G2/M (Figure 1a) and which persists in different mutants. The new experiments with controls by the side, support this hypothesis. We note that we performed the CHX experiments suggested by the reviewer, shown as Figure R1 for Reviewers only. Overall protein translation inhibition with CHX treatment leads to reduction of Mms4 and its phosphorylated version, but also causes delays in mitosis, likely because new protein synthesis is required for progression into a new cell cycle.

We are not sure why the interpretation of our Tc-HA-MMS4 experiment is difficult, but we imagine that it is a misunderstanding as our system is not a promotor shut-off, but a translation shut-off system where specifically Mms4 translation is affected upon addition of Tet. This approach allows to specifically prevent new Mms4 synthesis after G1 release, when we add Tet. In this way, in the absence of newly synthesized Mms4, it is possible to analyze how the levels of the protein (Mms4 and P-Mms4) are maintained throughout a whole new cell cycle. The experiments showed (Figure 1a) that Mms4 levels started to diminish in G2/M and had basically disappeared by the next G1; therefore, this was a useful approach for the further analyses throughout the paper.

In the revised version we included additional controls, including treatment versus no treatment with tetracycline (now Figure 1a), and performing identical experiments with *pADH1-MMS4* cells treated or not with tetracycline, to confirm that tetracycline alone does not have deleterious defects on Mms4 protein levels (Supplementary Fig. 1b).

4. Fig. 1, but all figures; information in the figure legends is too sparse and sometimes contradicting what is written in the text. For example: - Temperature of CDC48 experiment (just mentionend in text line 114-115); - Fig 1C: figure legends state that MG132 was added together with Tet after G1 release, text (line 101-102) states it was added 1h after release from G1

We improved the figure legends to provide the correct and complete information. Regarding the examples given by the reviewer, the description in the text was accurate.

5. Fig. 1b: The proteasome mutant seems to show much less phosphorylated MMS4 (if comparable with Fig 1a and the others), which is inconsistent with proteasomal degradation and the experiment with proteasome inhibition.

We have repeated the experiment to have them run in parallel with WT. In line with what we showed previously, the maximum level of phosphorylated Mms4 versus total Mms4 in the *cim3-1* mutant is lower than that of wt cells but persists throughout the cell cycle. This result does not invalidate the fact that P-Mms4 is stabilized in the mutant. In fact, we explain the result because both forms of Mms4 (P-Mms4 and non-P-Mms4) are being stabilized in the *cim3-1* proteasome mutant. We now provide this result in Figure 1c.

6. Fig. 1c: Phosphorylated form seems not very much stabilized by proteasome inhibition.

We repeated the MG132 experiment working in the *pdr5Δ* mutant to facilitate MG132 penetration through the yeast cell membrane. There is good stabilization of the Mms4 protein as also revealed in the quantifications. The experiment is shown in Figure 1b.

7. Fig. 1d: Again stabilization is not very visual. Is this just because of the use of different timepoints? I cannot judge whether the data is in support of the authors model, but if it was they are doing themselves a disfavor by the presentation.

We feel that the stabilization is quite visual especially in G1 and by looking at the ratio of phosphorylated Mms4 versus total. We repeated the experiment using also another mutant of Cdc48, *cdc48-6*, and present the results of the latter in the revised version as Supplementary Fig. 1d.

8. Fig. 2: In my eyes the clearest part of the paper. The stabilization of the phosphorylated version in G1 cells is believable. However, if they want to argue that all these factors act in one pathway, they need to show us double mutants and that those do not further increase the effect on degradation.

We have now further added quantifications that confirm the strong effects of the considered mutants on the stabilization of Mms4 phosphorylation. Moreover, in addition to the reported interactions between these ubiquitin ligases, the analyzed STUbL, Esc2 and Cul8 single mutants have similar effects in regard to deregulation of the nuclease activity of Mms4 (Figures 4-5) and on DNA damage response (Figure 6), suggesting that they act in the same pathway. The mutants of STUbL, Esc2 and Cul8 often show synthetic sick interactions between themselves, particularly *slx5Δ* in combination with *esc2Δ*, *cul8Δ*, *mms1Δ*, indicating that they have non-overlapping functions and substrates. In regard to their common substrate, Mms4, we could further analyze the dynamics in double *esc2Δ cul8Δ* and *esc2Δ mms1Δ* mutants, versus single mutants, which we now present in Supplementary Fig. 2b-c. We find similar stabilization of P-Mms4 with that observed in the single mutants.

9. Fig. 2a: An additional control using MMS4 without tag would be necessary.

We note that there is already a good control without His-tagged Smt3, in which we do not observe SUMOylated Mms4. However, we repeated the experiment to add an additional control using untagged Mms4.

10. Fig. 3a: They show enhanced levels of MMS4 on chromatin in the absence of SLX5 etc, but most likely this is simply due to enhanced levels of MMS4. Therefore, showing total levels of MMS4 in G1 will be meaningful.

We are now including whole cell extracts blots in the Supplementary Fig. 3a. In all mutants, as deduced from results presented above, there is persistent Mms4 phosphorylation in G1. As for the chromatin bound fractions, we included the Orc2 controls. The levels of Orc2 are similar among the considered strains, and therefore we are confident that equal amounts of chromatin fractions have been loaded in all cases. Thus, the higher levels of Mms4 on chromatin in the mutants is significant.

11. Fig. 4 and 5: To assess what is the function of the described degradation mechanism it will be necessary to find a specific mutant in MMS4 that is deficient in this degradative mechanism (for example: SUMO-site mutant, ubiquitination-site mutant, interaction-deficient point mutant). Otherwise, the authors have to use rather pleiotropic mutants.

This is a topic we have addressed intensively and now present the results in Figure 7 of the new manuscript. Starting from the fact that Esc2 interacts directly with Mus81-Mms4 (Sebesta et al, NAR, 2016), we used purified GST-Esc2 (versus GST control) to test interactions with peptide arrays of Mus81-Mms4 using the service of the PERperPRINT GmbH company in Heidelberg. Specifically, we used 15 amino acid (aa) Mms4- and Mus81-derived peptides with peptide-peptide overlaps of 14 aa. The assay with GST-Esc2 highlighted two main response peaks with significantly higher signal-to-noise ratios against peptides, one for Mms4 and one for Mus81, with the consensus motifs RSKKSSQVGKLGIKK (Mms4) and EKGTKKRKTRKYIPK (Mus81) (Supplementary Fig. 7a). Due to the significantly higher signal-to-noise ratios and the clear spot morphologies of the corresponding peptides, these two responses were considered specific.

We have next generated deletions to encode for small internal truncations in Mms4 ($\Delta 541-555$) and Mus81 ($\Delta 121-135$) and then combinations of these small truncations. The encoded Mus81-Mms4 complex has normal abundance and is still able to interact with Esc2 by co-IP (Supplementary Fig. 7b) but is partly defective in turnover and leads to stabilization of Mms4-P (Figure 7a). This mutant also reduces Rad53 phosphorylation (Figure 7b) and causes reduction in recombination structures associated with DNA replication (Figure 7c), resembling in these respects the results observed in *slx5 Δ* and *cul8 Δ* , *mms1 Δ* mutants (see Figures 5 and 6).

The rescue strategy with the promoter shut-off of MMS4 is not always convincing. There are comments on this below.

12. Fig. 4: The TC-MMS4 mutant relies on MMS4 degradation to work. Is it really a good strategy to use this mutant in the background of mutants that presumably are degradation deficient?

It is not an ideal solution, indeed, but the one we could use to bypass the synthetic lethality between *sgs1 Δ* and *mms4 Δ* mutations as well as synthetic lethal interactions between *sgs1 Δ* and *slx5 Δ* . Ideally Mms4 conditional inactivation is better suited to address the problem, as constitutive *mms4 Δ* may have other deleterious effects. We have made conditional alleles for *SGS1* and *MMS4* and identified viable combinations in which we can use *mms4 Δ* and *Tc-sgs1* and *slx5 Δ* whereby the rescue of the X-molecule loss associated with *slx5 Δ* (Figure 5a and Supplementary Fig. 5a) is suppressed by *mms4 Δ* . We present this new result in Figure 5a, together with another

independent experiment (Supplementary Fig. 5b). Because of space limitations, we are not presenting the initial experiment with *Tc-sgs1 slx5Δ Tc-MMS4*. The same is true for *cul8Δ* and *mms1Δ* mutations that decrease recombination structures in *Tc-sgs1-AID* (Supplementary Fig. 6a) and *Tc-sgs1* (Figure 5b and Supplementary Fig. 6b). *mms4Δ* mutation could suppress the loss in recombination structures associated with *mms1Δ* (Figures 5b) in similar trend with Tc-Mms4 (Supplementary Fig. 6b), where the rescue is best seen at late time points due to incomplete depletion, as pointed by the reviewer. Thus, we could strengthen the initial conclusions on the effects of Mms4 depletion (not ideal with Tc-Mms4 system and primarily visible at late time points) using the *mms4Δ* mutation (Figures 5a-b).

13. Fig. 4c: The authors show quantification of a single experiment and some of the effects appear to be small (and not easily visible on the gels itself; exception perhaps 240 min timepoint in Fig. 4b). How do we know they are meaningful/significant? Can the authors average over different experiments? Can the authors at least show that other experiments had the same trends?

As described at the point above, we now included separate results in different combinations with *sgs1* and *mms4* conditional alleles in Supplementary Figures 5 and 6.

14. Fig. 5a: How is the quantification done? How is it possible that Rad53 after 45min is only 35% phosphorylated. This number does not match the optical impression.

The quantifications were initially performed with the ImageJ software after acquiring digital pictures with the ChemiDoc Image Imaging System combined with the Image Lab Software from BIORAD. We are now using the corresponding software (Image Lab) for quantifications. The results match reasonably well the visual impression. We note that the provided quantifications are averaged for 3 blots from independent experiments, and we show one typical blot along the quantification graphs.

15. Fig. 5b: I am not convinced that the Mms4 over-expression scenario is analogous to the other mutant scenarios. How do the levels of Mus81 change if the authors inhibit degradation or overexpress Mms4.

16. Fig. 5b: Do the authors really want to concentrate purely on levels. Is it not more promising to focus on the presence of hyper-phosphorylated MMS4 G1 or S. In that vein, can they rescue this phenotype by using previously characterized phosphorylation-deficient mutants?

We agree that the experiment with *MMS4* overexpression is not fully analogous with the other mutant scenarios, as also pinpointed by other reviewers. Because it is not essential for the paper and we could add a stabilized Mus81-Mms4 mutant and its effect on Rad53 phosphorylation (Figure 7), we decided to remove this panel from the paper.

Reviewer #2 (Remarks to the Author):

'Mus81-Mms4 endonuclease is an Esc2-STUbL- 1 Cullin8 mitotic substrate tuning the DNA damage response'

In this study Waizenegger et al. report that phosphorylated Mms4 is channelled towards Cdc48 segregase-mediated proteasomal degradation during mitosis, in a process involving the SUMO-targeted Ubiquitin ligase Six5/8, Esc2 and Mms1-Cul8. Moreover they provide evidence suggesting that mitotic degradation of phosphorylated Mms4 avoids accumulation of active Mus81-Mms4 on chromatin during the subsequent S-phase, which would otherwise cause abnormal processing of recombination intermediates and delay activation of the checkpoint.

In my opinion, the work/model is very interesting and the findings merit publication in a broad journal such as Nature Communications. I would however suggest that the authors address the following aspects of their data, which in my opinion will help strengthen their model.

We are very happy that the reviewer finds our work of interest and worthy of publication in *Nature Communications*. We have addressed the reviewer's suggestions, which have been of great help to improve the manuscript.

Figure 1:

- The authors never formally show that the Tet-mediated inhibition of Mms4 translation actually works. To be able to claim it in the text, the authors should perform an experiment in which they omit Tet (or even better add Tet to a WT MMS4 strain). Otherwise it is unclear to me that the small decrease in Mms4 abundance at later timepoints of the cell cycle actually comes from the inhibition of translation

We have performed both experiments suggested by the reviewer and present them in Figure 1a and Supplementary Fig. 1b, respectively. The results show that Tet addition to a *Tc-HA-MMS4* strain strongly reduces Mms4 levels (both phosphorylated and non-phosphorylated forms, compared to the untreated *Tc-HA-MMS4* control (Figure 1a). Moreover, *pADH1-HA-MMS4* strain treated or not with Tet does not cause differences in Mms4 levels and dynamics of phosphorylation (Supplementary Fig. 1b).

- In Figure 1, but also in other Figures, Tubulin is clearly not a good loading control. It is obvious that its abundance is cell cycle regulated, which makes it difficult to then claim that levels of Mms4 fluctuate in a specific manner. Tubulin could be used if the authors loaded two conditions (e.g. +/- Tet) in the same gel – if in both timecourses tubulin looks identical, then Tet-mediated changes in Mms4 levels could be inferred.

In addition to loading the Tet and no Tet conditions side by side in the same gel, we have redone the experiments using Pgf1 as loading control. The conclusion remains the same, because the most important parameters are the ratios of phosphorylated Mms4 versus total Mms4 and of Mms4 levels versus the initial G1.

- Mms4 levels should be quantified, as previously done by the authors in other publications (e.g. Szakal 2013).

We have quantified the levels and present the quantifications close to the blots.

- Inhibition of the proteasome is known to delay cell cycle progression, for example by preventing the degradation of cyclins. Thus, the delayed degradation of Mms4 can be a consequence of cells getting stuck at G2/M. In essence, the authors should be careful

describing their data as it does not formally show that Mms4 is a proteasomal substrate.

To allow relatively normal cell cycle progression we used permissive conditions for the proteasome mutant (30°C instead of standard non-permissive 37°C). We have added the cell cycle profiles for each experiment to show their progression through the cell cycle and out of mitosis. We have also assessed the effect of inhibiting the proteasome with MG132 (Figure 1b). However, in line with the reviewer's suggestion, we have modified the text to say that Mms4 is ultimately degraded by the proteasome.

- All the findings in panels B, C and D could be explained if Cdc5 would not be efficiently degraded in proteasome or *cdc48* mutants. The authors should probe their samples for Cdc5. This is particularly important because Cdc5 is known to be a target of the APC/proteasome system.

We have done this and found persistence of Cdc5 in *cim3-1* and *cdc48* mutants (now presented in Figure 1c, Supplementary Fig. 1d). We note that the cells are not stuck in G2/M and reach G1 efficiently (see cell cycle profiles). While different explanations are indeed possible regarding the persistent Mms4 phosphorylation in G1 and we discuss the effect of persistent Cdc5 in this regard, we provide evidence in the paper for Mms4 turnover being driven via SUMO and ubiquitin modifications and interactions with Esc2. The maintenance of Mms4 phosphorylation in G1 is likely compounded by persistence of a small pool of Cdc5, the turnover of which seems to be linked with the one of Mms4 and mediated by the same factors.

- The authors describe Figure 1D as having normal kinetics of Mms4 phosphorylation. This is not the case as Mms4 is modified at least 15 minutes faster than in the WT.

We stated that the G2/M associated phosphorylation is observed, which is formally correct. It is true that Mms4 is modified "faster" than in wt, because *cdc48* cells show phosphorylated Mms4 already in G1 and this modification persists throughout the cell cycle. We have redone the experiment using a *cdc48-6* mutant and loading the WT and *cdc48-6* samples in the same gel (Supplementary Fig. 1d).

Figure 2:

- I am not sure the authors can formally claim that Mms4 is targeted by multiple Smt3 molecules. In panel A, the shifts could be caused by differential phosphorylation of sumoylated Mms4. The authors could soften their interpretation of the data or do an experiment where they phosphatase-treat the Ni-PD of Smt3.

The phosphatase treatment is not technically compatible with SUMO Ni-PD experiments performed in fully denaturing conditions. We softened the interpretation as suggested by the reviewer. The main message of this experiment is that Mms4 is SUMOylated. In regard to Mms4 being potentially targeted by SUMO chains, we are showing that SUMO chains are important for the normal cycle of Mms4 phosphorylation. Specifically, in conditions in which SUMO chains cannot be formed (i.e., in the *smt3-KRall* mutant), cells enter G1 with phosphorylated Mms4 (Figure 2c) and Mms4 monoSUMOylation is stabilized in Ni PDs (Figure 2d).

- Treatment with Nocodazole leads to a cell cycle arrest due to activation of the SAC, which is technically an M-phase arrest. Authors refer to it as G2, which is imprecise.

Indeed, Nocodazole treatment blocks cells in pro-metaphase. Because the terminology of M phase can give the idea of anaphase or telophase, we are now using G2/M

instead of G2 to refer to the nocodazole block and M instead of G2/M to refer to cells that have been released from the nocodazole arrest into mitosis.

- In panels B-E, authors should probe their WBs against Cdc5. It could easily be that Cdc5 is stabilised in the mutants analysed and lead to persistence of modified Mms4.

Indeed, we have done so and found stabilization of Cdc5 in the analyzed mutants. This suggests that STUbL and Cul8/Mms1 function on a small pool of Cdc5, likely associated with Mms4, as we found internal truncations in Mms4 and Mus81 with a similar effect on Mms4-P and Cdc5 stabilization (Figure 7a). We note that the persistence of Cdc5 in these mutants is compatible with exit from mitosis and that Mms4 degradation begins in G2/M when Cdc5 levels are at a peak (Figure 1a), compatible with the notion that Mms4 is the primary target for STUbL and Cullin8. The maintenance of Mms4 phosphorylation in G1 may indeed be associated with partial Cdc5 stabilization, which we now discuss in the paper.

- It is very obvious that Mms4 protein levels still decrease significantly in the mutants analysed. How do the authors explain this? It would be important to load the mutants analysed against the WT in the same gel and quantify Mms4 levels.

We have done so. Quantification of the phosphorylated form of Mms4 versus total Mms4 shows persistence of the phosphorylated form in the mutants, although the overall levels of Mms4 decrease. We infer that new Mms4 protein is synthesized at the end of mitosis or G1 in both WT and mutants.

Figure 3:

- Even though the data looks good, the Mus81-Mms4 activity assays could be quantified and the activity normalised to the protein in the IP.

We initially intended this result as qualitative. In wt cells, the nuclease activity is observed specifically in G2/M and there is no nuclease activity in G1. Whereas, we observe nuclease activity in *esc2Δ* cells arrested in G1 (and now we show the same for *cul8Δ* and *slx5Δ*). Nevertheless, we provide now quantification of the blots.

Figure 4:

- This data is overall very nice, but there is one aspect which I believe should be looked into. Even though Sgs1 levels are not quantified in the Wbs, which could be done, it seems to me that its depletion is not comparable in all conditions. My impression is that Sgs1 levels are always higher when joint molecules are lower, which, if true, would compromise some conclusion made. This is a critical experiment and I think authors should load all the conditions Vs. the WT and quantify carefully Sgs1 levels from the same gel. Otherwise the interpretation of the data could be ambiguous.

Generally, Sgs1 depletion works very well, although the Western blots to check the depletion can be challenging and non-quantitative because of the high molecular weight of the Sgs1 protein that causes variation in the efficiency of extraction in cell lysates. We reproducibly see similar effects of the mutants, as we emphasized in the revised version (see new Figure 5, Supplementary Figs. 5-6). We used the *Tc-sgs1* allele to bypass the conditional lethality of *sgs1Δ* with *slx5Δ* and *mms4Δ*, and the negative fitness interactions of *sgs1Δ* with *cul8Δ* and *mms1Δ*. We verified that the same is true using *cul8Δ* and *mms1Δ* in a *Tc-sgs1-AID* conditional strain in which Sgs1 depletion is induced by addition of Tetracycline and Auxin and is very efficient (Supplementary Fig. 6a), but still leads to a hypomorphic *sgs1* allele that causes synthetic sick interactions when combined with *slx5Δ*. Moreover, to verify and

strengthen the conclusions related to the role of Mms4 in the loss of recombination structures associated with STUbL and Cul8/Mms1 dysfunction, we have used *mms4* Δ in *Tc-sgs1 slx5* Δ and *Tc-sgs1 mms1* Δ and found reproducible rescue when Mms4 is simultaneously inactivated (Figure 5a, b and see Supplementary Fig. 5b).

Figure 5:

- The authors should treat WT and GAL-MMS4 cells with galactose and perform the same analysis. Comparing cells growing in two different carbon sources is not ideal as the growth rates and overall physiology of the cells will be different. I don't know it for certain, but I would not be surprised if the effect observed was due to the Galactose vs Raffinose rather than the over-expression of Mms4.

It is likely the case, as repetition with different carbon sources led to smaller differences. We decided to remove this panel, as there are several caveats associated with this experiment, as also pointed out by the other reviewers. Importantly, we obtained stabilized Mus81-Mms4 internal truncation mutants that show defects in Rad53 phosphorylation, which we now present in Figure 7b of the manuscript.

Figure 6:

- I like the first part of the model in panel A, but I don't think panel B is helpful. I don't think it summarises any of the important findings made in the paper.

We removed both panels due to space limitations caused by the new addition of Figure 7.

Reviewer #3 (Remarks to the Author):

This manuscript by Waizenegger et al describes an interesting analysis on Mms4 turn-over as required for proper chromosome segregation. The author's finding that MMS4 is affected by both Slx5/8 and Mms1/Cul ubiquitin ligases is interesting and new. Mms4 and Mms4-P were stabilized in *slx5Δ*, *esc2Δ*, and *cul8Δ* strains. Following up on earlier suggestive data, the authors confirm that the Mms4 and Mms4-P are degraded in a cell-cycle dependent manner. This degradation can be prevented (or reduced) when proteasome activity is prevented. Additionally, Mms4 and Mms4-P are stabilized in a *cdc48* mutant which apparently also fails to extract Mms4 from chromatin.

We are happy the reviewer finds our results of interest and novel.

Proof that Mms4 is sumoylated in Fig 2a is marginal and ubiquitylation was not shown directly.

We do not think that proof on SUMOylation is marginal as Ni PD of SUMOylated species is usually the method of choice to drive such a point. We also show that SUMO chains are involved in Mms4-P turnover and defects in forming SUMO chains stabilize monoSUMOylated forms of Mms4 species (Figures 2c-d).

We agree that the ubiquitylation part was speculative and not directly supported by experimental evidence. We have now performed Ni-PD of ubiquitylated species in cells expressing His-tagged ubiquitin. We observe that Mms4 is ubiquitylated and that this ubiquitylation is reduced in *cul8Δ* mutants, a new result we added to the paper (Figure 3c).

I am not clear how the results in Fig. 4 reveal a role for Slx5/8 and Mms1/Cul ubiquitin ligases in replication-associated recombination intermediate stability by preventing abnormal Mms4-Mus81 action associated with defective turnover. The authors need to either discuss the data in more detail or provide additional data to drive this point home.

We worked hard to strengthen this result and provide new results with tighter alleles and *mms4Δ*. The 2D gels revealed a reduction in the X-shaped intermediates (highlighted with arrows in the gels) in *mms1Δ*, *cul8Δ* and *slx5Δ* mutants, thus uncovering a role for these factors in the formation/stability of replication-associated recombination structures. We used initially the *Tc-sgs1* allele to bypass the conditional lethality of *sgs1Δ* with *slx5Δ* and *mms4Δ*, and the negative fitness interactions of *sgs1Δ* with *cul8Δ* and *mms1Δ*. We verified that the conclusion holds using *cul8Δ* and *mms1Δ* in a *Tc-sgs1-AID* conditional strain in which Sgs1 depletion is induced by addition of Tetracycline and Auxin and is very efficient (Supplementary Fig. 6a), but still causes a hypomorphic *sgs1* allele that leads to synthetic sick interactions when combined with *slx5Δ*. Moreover, to verify and strengthen the conclusions related to the role of Mms4 in the loss of recombination structures associated with STUbL (Slx5) and Cul8/Mms1 dysfunction, we used *mms4Δ* in *Tc-sgs1 slx5Δ* and *Tc-sgs1 mms1Δ* backgrounds and found reproducible rescue when Mms4 is simultaneously inactivated. We are showing these new results and revisited the text to ensure clarity in the experimental description.

Finally, data on Rad9 and Rad53 phosphorylation suggests that DNA damage checkpoints are not properly engaged if MMS4 turn-over is defective. However, differences in Rad9 and Rad53 phosphorylation between WT and mutants seem minimal (at least by visual inspections) and there are questions how the quantitation was performed.

We feel that the differences are visual and well supported by quantifications, which we have redone using the Image Lab software. We note that the provided quantifications are averaged for 3 blots from independent experiments, and we show one typical blot along the quantification graphs.

Overall, this is a well-written and interesting story that seeks to establish the interplay of PTMs of Mms4 and protein degradation machinery to prevent its inappropriate activity in the G1 stage of the cell cycle.

We are happy the reviewer finds our paper well written and interesting story.

MAJOR:

Proof that MMS4 is sumoylated in Fig 2a is marginal. MMS4-PK was co-expressed with SMT3-HIS6. After ppt of MMS4PK the authors detect SUMO-HIS6 in the pull-down lanes. This could be improved with a SUMO-shift assay with epitope-tagged and untagged SUMO. Likewise, ubiquitylation of Mms4 in WT and ubiquitin ligase mutants should be shown in the main document.

As mentioned above, we do not agree on the point that proof on Mms4 SUMOylation is marginal. We have performed Ni PD of SUMOylated species in cells expressing His-tagged SUMO and either Mms4-PK (Figure 2a) or HA-Mms4 (Figure 2d) in different conditions of culture and mutants and with appropriate controls (i.e. cells not expressing His-tag SUMO and even additionally non-tagged strains). We also show that SUMO chains are involved in Mms4 turnover linked to the phosphorylation cycle and defects in forming SUMO chains stabilize monoSUMOylated forms of Mms4 species and Mms4-P (now Figures 2c-d). SUMO-shift assay could enhance this point further, but we feel that there is sufficient evidence for the SUMOylation concept. The ubiquitylation part was not demonstrated in the paper being just an inference. To address this weakness, we established HIS-tagged-Ubi4 under the control of *GAL* promoter, and found that Mms4 is ubiquitylated, with this modification being reduced in *cul8Δ* mutants. We added this new result in Figure 3c.

Fig 5: This is a western blot film that appears to be scanned for quantitation. Since the authors are quantitating minute differences in protein levels and because western blot signals on film are non-linear, this technique is not appropriate for data quantitation. The authors should use an Odyssey or similar system to quantitate their western blots. The description how the data was analyzed is at a minimum (e.g. what were the bands normalized to?). Additionally, taking into account that levels of the loading control as well as the G1 levels vary, there is reduced confidence in these data. Finally, WT and *slx5Δ* (and possibly other strains) show vastly different growth rates and so a comparison of checkpoint protein activation is difficult at best.

Fig 5b quantitation has the same problems with the additional caveat that *GAL* overexpression phenotypes are very difficult to interpret.

The quantifications were initially performed with the ImageJ software after acquiring digital pictures with the ChemiDoc Image Imaging System combined with the Image Lab Software from BIORAD. We are now using the corresponding software (Image Lab) for quantifications. The results match reasonably well the visual impression and the results are not minute as they show statistical significance. Although *slx5Δ* and to some extent *cul8Δ*, *mms1Δ* and *esc2Δ* cells have increased doubling times compared to WT, exponentially growing overnight cultures of these mutants can be arrested well in G1 and progress similarly with WT in for the one cell cycle relevant to our experiment. Therefore, we have confidence in these results.

We agree that the *GAL-MMS4* overexpression experiment initially presented in Figure 5b had several caveats also pinpointed by the other reviewers, and removed it. Importantly, we managed to identify an *Mms4-Mus81* mutant with increased *Mms4-P* stability (Figure 7a). This mutant has normal growth rate and cell cycle progression, but shows defects on *Rad53* phosphorylation, which we present in Figure 7b.

MINOR:

In Fig 1a (Tc-HA-MMS4(WT) ... there seems to be another prominent band (partially cut of) in the lane that runs above MMS4-P. The authors should show more of the blot and comment on the identity of this band.

It is a non-specific band, we are showing all the uncropped blots as Supplementary Data and labeled this band as “unspecific”.

- line 160-161 ... the authors indicate that MMS4-P is stabilized ... but so is the non P form of MMS4! This should be noted and discussed.

Yes, indeed, both forms of *Mms4* are degraded in mitosis (Figure1) and *Mms4-P* is stabilized in the *STUbL*, *Esc2* and *Cullin8* mutants (Figures 2 and 3). We paid attention to clarify this point through the manuscript.

- The role of SUMO in degradation of MMS4 is unclear as both MMS4 and MMS4-P are degraded and stabilized. The authors should discuss this.

Yes, we are discussing this. *Mms4* and *Mms4-P* are SUMOylated and this modification can be extended to SUMO chains that attract *STUbL-Esc2* and *Esc2/Cul8/Mms1* causing additional ubiquitylation and proteasomal degradation of the substrate. In addition, a small pool of *Cdc5*, likely associated with *Mms4*, is regulated via the *STUbL-Esc2-Cul8* axis, a point we may develop further in a future work.

- Fig 3: Enhanced chromatin association of MM4 in *esc2*, *slx5Δ cul8Δ* is beautiful. I am wondering if tubulin is the best control though.

As basically all *Mms4* is bound to chromatin, *Orc2* is the right control for the chromatin fractions. *Orc2* shows similar amounts in all strains, so the *Mms4* differences found among them are significant. Tubulin is a good control for the soluble proteins, and as these are about 95% of the WCE, it is a good control to estimate that similar amounts of total proteins were loaded. Nevertheless, we are now including a typical blot of total WCEs in G1 as Supplementary Fig. 3a.

- Line 182: Thus, defective *Mms4-P* turnover causes persistent high activity of the *Mus81-Mms4* nuclease on chromatin. ... this was only shown for *esc2Δ* (3b) ... does it behave the same in *slx5Δ* or *cul8Δ*? This is important to assess the role that sumoylation and *STUbLs* play in this process.

We have analyzed and report now that the same happens in *slx5* and *cul8* mutants (Figure 4b).

... is the figure mislabeled? Both left and right figures are labeled Tc-HA-MMS4 WT *esc2Δ*

It was not mislabeled. We showed assays with WT and *esc2Δ* strains in both panels, with different DNA substrates: left, a replication fork; right, a 3'-flap. To avoid confusion,

we have now modified the figure to show only a 3'-flap substrate for all mutants (Figure 4b).

- Line: 205 cells depleted for Sgs1 accumulated X-shaped intermediates ... please indicate (e.g. using an arrow) these features. What is being quantitated here?

We have labeled the figure better and indicated with arrows the signal that is being quantified.

- Along the same line, I am not clear how the results in Fig. 4 reveal a role for MMS1, CUL8 and SLX5 in replication-associated recombination intermediate stability by preventing abnormal Mms4-Mus81 action associated with defective turnover. The authors need to either discuss the data in more detail or provide additional data (and controls) to drive home this point.

We have commented on this point above and added several new results in Figures 5 and Supplementary Fig. 5-6.

Line 255: We found that Mms4-P is engaged by posttranslational modifications with SUMO and ubiquitin ... BUT no ubiquitin is shown in main figures of the ms.

We have now added evidence also on ubiquitylation, as commented above.

Line 266: Extensive phosphorylation of Mus81-Mms4 may cause its unfolding, exposing buried lysine residues leading to multi, poly-SUMOylation ... Unfolding is very unlikely, maybe just state "make accessible"?

We modified the text accordingly.

Figure R1

Time course experiment to analyze Mms4 levels and species in a cycloheximide chase experiment. Logarithmically (log) grown Mms4-PK cells were synchronized in G1 phase with α -factor and then released in YPD medium to reach G2/M. At this point, 50 mg/ml cycloheximide (CHX) or only DMSO (ctrl) was added to the cells and samples were taken at the indicated timepoints. The presence of PK-tagged unphosphorylated (Mms4^{PK}) or phosphorylated Mms4 (P-Mms4^{PK}) was analyzed by Western Blot. Pgk1 served as a loading control. Total levels of Mms4 were quantified by normalization to the loading control and are shown relative to the G1 phase sample.

REVIEWERS' COMMENTS

Reviewer #1 (Remarks to the Author):

Comments to authors:

The revised version of the manuscript "Mus81-Mms4 endonuclease is an Esc2-STUbL-Cullin8 mitotic substrate tuning the DNA damage response" by Dana Branzei and coworkers is an impressive improvement compared to the previous version and addresses all points previously raised. In particular, the mutant version of Mus81-Mms4 that stabilized the nuclease complex and the new figure 7 provides a "flagstone" for this paper and shows that defects of yeast cells deficient in the Esc2-STUbL-Cullin8 axis in protecting replication/recombination intermediates are indeed linked to deregulated Mus81-Mms4. I can therefore fully recommend publication in Nature Communications.

The authors may consider to provide their thoughts on the following topics, which I think readers may find insightful.

1 – All mutants deficient in the Esc2-STUbL-Cullin8 axis lead to Mms4 stabilization. Yet, only mutants in Cul8 lead to defects in ubiquitination. How do the authors envision such a dual ubiquitin ligase works, in particular the STUbL? Do they know of other examples?

2 – The mutations used in Figure 7 lead to stabilization, but Esc2 interaction by CoIP is still there. Can the author speculate about the nature of their mutant? Does it lower affinity to Esc2 or does it work in a different/unexpected way?

3 – The authors suggest that Mus81-Mms4 degradation is linked to degradation of a sub-pool of Cdc5. Can they speculate what is the nature of this sub-pool?

Reviewer #2 (Remarks to the Author):

The authors have done a thorough revision of their work and improved it significantly. I think that the main messages are supported by the data. A couple of minor issues I think should/could still be addressed are the following:

1- In figure 5, the authors use an HA-tagged version of Sgs1. In very recent work, the Lichten lab has shown that HA-tagged Sgs1 isn't properly functional and exacerbates other combinatorial phenotypes, making the interpretation of data rather complicated (Cohen and Lichten 2020). It might be desirable to make sure that this isn't the case here.

2- The Mus81-Mms4 mutant described in Figure 7 is very interesting, but I am not sure what to think about it. First, it still binds Esc2, while the prediction is that it shouldn't. Second, Cdc5 levels are very stable in this background. The authors claim that it is a very small proportion of Cdc5, but it is a very significant part (by eye, ~50%). This is difficult to interpret, in my opinion. I don't doubt the data, I am just very surprised about this result. I think that the authors should probe for other Cdc5 targets in some of their WBs (e.g. Scc1, etc), as well as for other proteins that follow a pattern that is similar to Cdc5 (e.g. Clb1, Clb2, etc). It would be good to address the specificity of the inter-connected levels of phosphorylated Cdc5 and Mms4.

3- The authors show nicely that Mms4 is stabilised in $slx5\Delta$, etc, and in proteasome mutants/with proteasome inhibitors. However, I would argue that there is no definitive data to say that the proteasome directly degrades Mms4. In some sentences in the text it feels that it is certain that the proteasome degrades Mms4 (e.g. line 112). The proteasome (and Slx5, etc) target hundreds of different things, which could (potentially) indirectly lead to the results observed.

Reviewer #3 (Remarks to the Author):

The authors did a good job addressing the reviewer's comments and the manuscript is much approved.

Oliver Kerscher, Ph.D.

Reviewers' comments:

Reviewer #1 (Remarks to the Author):

Comments to authors:

The revised version of the manuscript “Mus81-Mms4 endonuclease is an Esc2-STUbL-Cullin8 mitotic substrate tuning the DNA damage response” by Dana Branzei and coworkers is an impressive improvement compared to the previous version and addresses all points previously raised. In particular, the mutant version of Mus81-Mms4 that stabilized the nuclease complex and the new figure 7 provides a “flagstone” for this paper and shows that defects of yeast cells deficient in the Esc2-STUbL-Cullin8 axis in protecting replication/recombination intermediates are indeed linked to deregulated Mus81-Mms4. I can therefore fully recommend publication in Nature Communications.

We are happy that the reviewer appreciates the intense effort that went into revision and now fully recommends publication of our manuscript in Nature Communications.

The authors may consider to provide their thoughts on the following topics, which I think readers may find insightful.

1 – All mutants deficient in the Esc2-STUbL-Cullin8 axis lead to Mms4 stabilization. Yet, only mutants in Cul8 lead to defects in ubiquitination. How do the authors envision such a dual ubiquitin ligase works, in particular the STUbL? Do they know of other examples?

We would like to note that our data do not rule out that STUbL ubiquitylates polySUMOylated Mms4, but detection of such SUMO-Ubiquitin hybrid species is technically very challenging as they are prone to accumulate in the wells (Psakhye et al, Mol Cell, 2019). Thus, STUbL and Cullin8 may separately affect Mms4 turnover and cause stabilization of phosphorylated Mms4 species via different mechanisms. We are now commenting about this point in the manuscript. We do not know at the moment of other examples of shared substrates between STUbL and Cullin8. One potential connection between these ubiquitin ligases in terms of substrate preference could be directed by Esc2. It will be useful in the future to systematically assess shared substrates, starting from reported STUbL substrates (Psakhye et al, Mol Cell, 2019; Albuquerque et al, PLoS Genetics, 2013) or other factors whom we found to persist in all three mutants, such as Cdc5. However, at this point in time, we do not feel that speculation on shared substrates of STUbL and Cullin8 and their mode of action is possible.

2 – The mutations used in Figure 7 lead to stabilization, but Esc2 interaction by CoIP is still there. Can the author speculate about the nature of their mutant? Does it lower affinity to Esc2 or does it work in a different/unexpected way?

In the mutant Mms4-Mus81 with stabilized Mms4 phosphorylation we truncated patches of amino acids with strongest interaction signal for Esc2 in the peptide scan approach, but we did not lose the physical interaction by co-IP. As there are additional interaction modules identified in Mms4-Mus81 (Supplementary Fig. 8a), which may become critical in physiological conditions when the endonuclease is bound to chromatin, we cannot make an accurate prediction on whether the interaction affinity is lowered. We suggested this possibility in the text, but we will need other methodologies to probe this hypothesis.

3 – The authors suggest that Mus81-Mms4 degradation is linked to degradation of a sub-pool of Cdc5. Can they speculate what is the nature of this sub-pool?

Cdc5 is known to physically engage its targets via its Polo-box domain (Almawi et al, *Sci Rep* 2020). In a simplified view, it is possible that a fraction of Cdc5 associated with Mus81-Mms4 undergoes degradation at the same time with the endonuclease.

Reviewer #2 (Remarks to the Author):

The authors have done a thorough revision of their work and improved it significantly. I think that the main messages are supported by the data. A couple of minor issues I think should/could still be addressed are the following:

We are happy that the reviewer appreciates the revised version. We are answering to the reviewer's points below.

1- In figure 5, the authors use an HA-tagged version of Sgs1. In very recent work, the Lichten lab has shown that HA-tagged Sgs1 isn't properly functional and exacerbates other combinatorial phenotypes, making the interpretation of data rather complicated (Cohen and Lichten 2020). It might be desirable to make sure that this isn't the case here.

Indeed, the work of Cohen and Lichten, G3, 2020 reported that C-terminal tagging of Sgs1, particularly with a 6HA tag, affects its function. Here we used N-terminal tagging of Sgs1 with 3HA. Moreover, we used the mutant Tc-HA-Sgs1 in a conditional manner, to deplete Sgs1 at a specific point in time, rather than study its function per se.

2- The Mus81-Mms4 mutant described in Figure 7 is very interesting, but I am not sure what to think about it. First, it still binds Esc2, while the prediction is that it shouldn't. Second, Cdc5 levels are very stable in this background. The authors claim that it is a very small proportion of Cdc5, but it is a very significant part (by eye, ~50%). This is difficult to interpret, in my opinion. I don't doubt the data, I am just very surprised about this result. I think that the authors should probe for other Cdc5 targets in some of their WBs (e.g. Scc1, etc), as well as for other proteins that follow a pattern that is similar to Cdc5 (e.g. Clb1, Clb2, etc). It would be good to address the specificity of the inter-connected levels of phosphorylated Cdc5 and Mms4.

In the mutant Mms4-Mus81 we truncated patches of amino acids with strongest interaction signal for Esc2 in the peptide scan approach, but we did not abolish the physical interaction as detected by co-IP. As there are additional Esc2-interaction modules identified in Mms4-Mus81 by the peptide scan approach (Supplementary Fig. 8a), which may be critical in physiological conditions, we cannot make an accurate prediction on whether the interaction affinity is lowered. We suggested this possibility in the text, but we will need other methodologies to probe this hypothesis.

For the second point, we observe stabilization of Cdc5 and Clb2 in mutants where Mus81-Mms4 is stabilized (Fig. R1), in conditions in which flow cytometry analysis reveals a rather normal progression of cells through mitosis (See Fig. 7a and Fig. R1). The same is true in *esc2*, *slx5* and *cul8* mutants in which there is persistence of both Cdc5 and Clb2 (Figure R2). Thus, several degradation processes occurring in mitosis are impaired in *esc2*, *slx5* and *cul8* mutants, but other studies will be needed in the future to address mechanistically these observations and possibly reveal the links between STUbL, Cullin8 and the APC/C degradation system in mitosis.

3- The authors show nicely that Mms4 is stabilised in *slx5Δ*, etc, and in proteasome mutants/with proteasome inhibitors. However, I would argue that there is no definitive data to say that the proteasome directly degrades Mms4. In some sentences in the text it feels that it is certain that the proteasome degrades Mms4 (e.g. line 112). The proteasome (and Slx5, etc) target hundreds of different things, which could (potentially) indirectly lead to the results observed.

We have revised the text to express the notion that Mms4 is likely targeted for proteasome-mediated degradation.

Reviewer #3 (Remarks to the Author):

The authors did a good job addressing the reviewer's comments and the manuscript is much approved.

We are happy that the reviewer full approves our manuscript for publication.

Fig. R1

Fig. R2